

# Biological weathering and its consequences at different spatial levels – from nanoscale to global scale

Roger D. Finlay[1], Shahid Mahmood[1], Nicholas Rosenstock[2], Emile B. Bolou-Bi[3], Stephan J. Köhler[4], Zaenab Fahad[1,5], Anna Rosling[5], Håkan Wallander[2], Salim Belyazid[6], Kevin Bishop[4] and Bin Lian[7]

[1]Uppsala BioCenter, Department of Forest Mycology and Plant Pathology, Swedish University of Agricultural Sciences, SE-750 07 Uppsala, Sweden
[2]Department of Biology, Lund University, Box 117, SE-221 00 Lund, Sweden
[3]Université Felix Houphouët-Boigny, UFR des Sciences de la Terre et des Ressources Minières, Departement des Sciences du sol, BP 582 Abidjan 22, Côte D'Ivoire
[4]Soil-Water-Environment Center, Department of Aquatic Sciences and Assessment, Swedish University of Agricultural Sciences, SE-750 07 Uppsala, Sweden
[5]Department of Ecology and Genetics, EBC, Uppsala University SE-752 36 Uppsala, Sweden
[6]Department of Physical Geography, Stockholm University, SE-106 91 Stockholm, Sweden
[7]Laboratory College of Life Sciences, Nanjing Normal University, Nanjing 210023, China

*Correspondence to*: Roger D. Finlay (Roger.Finlay@slu.se)

**Abstract.** Plant nutrients can be recycled through microbial decomposition of organic matter but replacement of base cations and phosphorus, lost through harvesting of biomass/biofuels or leaching, requires *de novo* supply of fresh nutrients released through weathering of soil parent material (minerals and rocks). Weathering involves physical and chemical processes that are modified by biological activity of plants, microorganisms and animals. This article reviews recent progress made in understanding biological processes contributing to weathering. A perspective of increasing spatial scale is adopted, examining the consequences of biological activity for weathering from nanoscale interactions, through *in vitro* and *in planta* microcosm and mesocosm studies, to field experiments and finally, ecosystem and global level effects. The topics discussed include: the physical alteration of minerals and mineral surfaces, the composition, amounts, chemical properties and effects of plant and microbial secretions, and the role of carbon flow (including stabilization/sequestration of C in organic and inorganic forms). Although the predominant focus is on the effects of fungi in forest ecosystems, the properties of biofilms, including bacterial interactions, are discussed. The implications of these biological processes for modelling are discussed and finally, we attempt to identify some key questions and knowledge gaps, as well as experimental approaches and areas of research in which future studies are likely to yield useful results. A particular focus of this article is to improve the representation of the ways in which biological processes complement physical and chemical processes that mobilize mineral elements, making them available for plant uptake. This is necessary to produce better estimates of weathering that are necessary for sustainable management of forests in a post-fossil fuel economy. While there are abundant examples of nm- and μm-scale physical interactions between microorganisms and different minerals, opinion appears to be divided with respect to the quantitative significance of these observations to overall weathering. Numerous *in vitro* experiments and microcosm studies involving plants and their associated microorganisms suggest that the allocation of plant-derived carbon, mineral dissolution and plant nutrient status are tightly





coupled but there is still disagreement about the extent to which these processes contribute to field-scale observations. Apart from providing dynamically responsive pathways for the allocation of plant-derived carbon to power dissolution of minerals, mycorrhizal mycelia provide conduits for the long-distance transportation of weathering products back to plants that are also quantitatively significant sinks for released nutrients. These mycelial pathways bridge heterogeneous substrates, reducing the

influence of local variation in C:N ratios. The production of polysaccharide matrices by biofilms of interacting bacteria and/or fungi at interfaces with mineral surfaces and roots, influences patterns of production of antibiotics and quorum sensing molecules, with concomitant effects on microbial community structure, and the qualitative and quantitative composition of mineral solubilizing compounds and weathering products. Patterns of carbon allocation and nutrient mobilization from both organic and inorganic substrates have been studied at larger spatial and temporal scales, including both ecosystem and global

levels and there is a generally wider degree of acceptance of the 'systemic' effects of microorganisms on patterns of nutrient mobilization. Theories about the evolutionary development of weathering processes have been advanced but there is still a lack of information connecting processes at different spatial scales and detailed studies of the liquid chemistry of local weathering sites at the µm-scale, together with up-scaling to soil-scale dissolution rates, are advocated, as well as new approaches involving stable isotopes.

**1 Introduction**

Modelling of base cation supply using the PROFILE/FORSAFE modelling platform (Sverdrup and Warfvinge, 1993; Wallman et al., 2005) suggests that planned intensification of Swedish forestry, involving increased harvesting of organic residues for biofuel, will not be sustainable in a long-term perspective without compensatory measures such as wood ash recycling (Akselsson et al., 2007; Klaminder et al., 2011; Futter et al., 2012; Moldan et al., 2017). The base cations and phosphorus that

are essential for forest growth can be re-cycled from organic residues through microbial decomposition but if they are lost through removal of organic material the only way they can be replaced is by weathering of rocks and minerals. There is a need to improve the available estimates of weathering and to improve our knowledge of the ways in which biological processes may complement physical and chemical processes that mobilize mineral elements, making them available for plant uptake.

The role of fungi in biological weathering in boreal forest soils was reviewed by Hoffland et al. in 2004 and by Finlay et

al. in 2009. More recent reviews of the more specific roles of mycorrhizal symbiosis in mineral weathering and nutrient mining from soil parent material (Smits and Wallander, 2017), pedogenesis (Leake and Read, 2017) and immobilization of carbon in mycorrhizal mycelial biomass and secretions (Finlay and Clemmensen, 2017) have also been published. Twelve testable hypotheses on the geobiology of weathering were outlined by Brantley et al. (2011). These authors concede that some of the outlined hypotheses have been implicit in scientific research conducted since the late 1800's but argue that there are now new

analytical, modelling and field opportunities to test these hypotheses. The aim of the present article is to review advances in the understanding of biological weathering that have been made since 2009, particularly with respect to nutrient and carbon cycling within boreal forests, including findings made within the interdisciplinary project *Quantifying Weathering Rates for*



*Sustainable Forestry* (QWARTS, 2012-2016) funded by FORMAS, the Swedish Research Council for Environment, Agricultural Sciences and Spatial Planning. One major motivation for this study was the concern that the modelling tools used to determine the long-term supply of weathering products for sustainable forest growth may have been missing biological processes that allow a forest ecosystem to alter the rate of weathering in response to the biological demand for these weathering

products (Klaminder et al., 2011).

Biological (or biogenic) weathering involves the weakening and disintegration of rocks and dissolution of minerals, caused by the activity of plants, animals and microorganisms. Biological weathering takes place in conjunction with physical and chemical processes but there is still disagreement about the quantitative contribution of biogenic weathering to overall weathering (see Finlay and Clemmensen, 2017; Leake and Read, 2017; Smits and Wallander, 2017). The supply and transport

of photosynthetically derived carbon through roots and mycorrhizal hyphae to organic and inorganic substrates is a fundamental biogeochemical process (Jones et al., 2009), influencing both decomposition and mineral weathering, and these two processes influence each other (Fig. 1). This flow of carbon and the role of plant-microbe-soil interactions in the rhizosphere have been reviewed in an evolutionary perspective (Lambers et al., 2009) and with respect to their potential applications in sustainable agriculture, nature conservation, the development of bio-energy crops and the mitigation of climate

change (Philippot et al., 2013). Since there is disagreement about whether biological processes demonstrated at small spatial scales contribute significantly to field scale processes we have adopted a spatial perspective and review processes occurring at the nm- and μm-scale before discussing *in vitro* microcosm experiments, mesocosm studies with plants, field experiments and finally studies of effects at the ecosystem and global scale.

## 2 Microscale/nanoscale observations of physical alteration of minerals

The idea that microorganisms may alter rocks and minerals is not new and biogenic etching of microfractures in borosilicate glass and crystalline silicates (olivine) by microfungi (*Penicillium notatum* and *Aspergillus amstellodami*), presumed to be producing both organic acids and siderophores, was demonstrated by Callot et al. in 1987. Early studies by Paris et al. (1995, 1996) demonstrated *in vitro* weathering of phlogopite involving displacement of non-exchangeable interlayer $K^+$ and alteration of the crystal lattice structure, as well as stimulated accumulation of oxalate under simultaneous $K^+$ and $Mg^{2+}$ deficiency. The

widespread occurrence of tubular pores, 3-10 μm in diameter (Fig. 2), has been demonstrated in weatherable minerals in podzol surface soils and shallow granitic rock under European coniferous forests (van Breemen et al., 2000; Jongmans et al., 1997; Landeweert et al., 2001). Some of these pores were found to be occupied by fungal hyphae and the authors speculated that they could be formed by the weathering action of hyphae (possibly in association with bacteria) releasing organic acids and siderophores. The aetiology of pore formation has been questioned however, with some authors claiming that the observed

pores are of abiotic origin (Sverdrup, 2009). Studies of feldspar tunneling along chronosequences created by post-glacial rebound (Hoffland et al., 2002) revealed that the tunnels were more frequent in the uppermost 2 cm of the E horizon, that the frequency of tunneling increased with soil age, and that there was a lag period of up to 2000 years when tunnels were absent





or rare, postulated by the authors to coincide with the time taken for the disappearance of the more easily weatherable K and Ca containing biotite and hornblende. Parallel studies along productivity gradients (Hoffland et al., 2003) have also revealed a significant positive correlation between the density of ectomycorrhizal root tips and the density of tunnels in the E horizon. However similar tunnels in feldspars across a sand dune chronosequence at Lake Michigan have been estimated to contribute

less than 0.5% of total mineral weathering (Smits et al., 2005) suggesting either that fungal weathering is negligible, or that tunnel formation reflects only a small proportion of the total weathering effect of the fungi. The total mineral surface area available for mineral weathering in most mineral soils is clearly much larger than the internal surface area of the observed tunnels and small tunnel-like features were observed in mineral surfaces by Smits et al. (2005). Different biomechanical mechanisms used by fungi to penetrate rock have received increasing attention (Sterflinger, 2000). Ultramicroscopic and

spectroscopic observations of fungus-biotite interfaces during weathering of biotite flakes have revealed biomechanical forcing, and altered interlayer spacing associated with depletion of K by an ectomycorrhizal fungus (*Paxillus involutus*) (Bonneville et al., 2009). It appears that physical distortion of the lattice structure takes place before chemical alteration through dissolution and oxidation. Fungal hyphae colonizing fractures and voids in minerals can exert substantial mechanical force and have been demonstrated to build up turgor pressures in excess of 8 MPa that is sufficient to penetrate Mylar and

Kevlar and widen existing cracks in rocks (Howard et al., 1991). Recent studies of biotite colonization by *P. involutus* (Bonneville et al., 2016) have revealed extensive oxidation of Fe(II) up to 2 μm in depth and the increase in Fe(III) implies a volumetric change that is sufficient to strain the crystal lattice and induce the formation of microcracks, which are abundant below the hypha-biotite interface.

      Since the first observations of Jongmans et al. (1997), stimulated interest in biogenic weathering has led to a large number

of subsequent studies. The endolithic biosignatures of rock inhabiting microorganisms can be distinguished from purely abiotic microtunnels (McLoughlin et al., 2010). Biological tubular microcavities can be distinguished by their shapes, distribution, and the absence of intersections which excludes an origin by chemical dissolution of pre-existing heterogeneities such as, radiation damage trails, gas-escape structures, or fluid inclusion trails. Atomic force microscopy and scanning transmission electron microscopy-energy dispersive X-ray spectroscopy (STEM-EDX) have been used to demonstrate nanoscale alteration

of surface topography and attachment and deposition of organic biolayers by fungal hyphae (Bonneville et al., 2011; McMaster, 2012; Gazzè et al., 2013, 2014; Saccone et al., 2012). The data from these studies suggest that bio-mechanical forcing takes place with μm-scale acidification mediated by surface-bound hyphae and subsequent removal of chemical elements due to fungal action. However, so far, the quantitative significance of these effects to total weathering rates is still unclear. Comparative studies of forests with either ectomycorrhizal or arbuscular mycorrhizal host tree species (Koele et al.,

2014) have revealed the presence of tunnel like structures in minerals in both types of forest, suggesting that mineral weathering can be caused by acidification of the rhizosphere by both types of mycorrhizal fungus and/or saprotrophic fungi. Investigations of silicate mineral surfaces, buried in proximity to roots of trees that would normally host arbuscular mycorrhizal fungi and were growing in an arboretum (Quirk et al., 2012, 2014), suggest that arbuscular mycorrhizal fungi may also form weathering trenches, although the associated fungi were not identified in these particular studies. Nanoscale channels in chlorite flakes



colonized by ectomycorrhizal fungi have also been demonstrated (Gazzè et al., 2012) using atomic force microscopy, and suggested as evidence that fungal activity, fuelled by plant photosynthate, can enhance mineral dissolution.

### 3 Biofilms and small scale microbial interactions with consequences at higher spatial scales

Most microorganisms do not live as pure cultures of dispersed single cells in soil solution. Instead they aggregate at interfaces – on surfaces of roots, organic matter, rocks and minerals, forming biofilms or microbial mats (Flemming and Wingender, 2010; Flemming et al., 2016). Biofilms consist of a hydrated matrix of extracellular polymeric substances (EPS), mostly produced by the organisms they contain. This matrix can account for 90% of the dry mass of the biofilm and provides a structural scaffold responsible for adhesion to surfaces and cohesion of the biofilm, enabling interactions that are entirely different from those of planktonic bacteria. The EPS matrix isolates microorganisms from the bulk soil solution, maintaining them in close proximity to each other and to substrate surfaces, concentrating weathering agents and allowing cell to cell communication and quorum sensing by containing and concentrating signal molecules. This permits the formation of synergistic microbial consortia, production, accumulation, retention and stabilization of extracellular enzymes through binding interactions with polysaccharides, sorption of organic compounds and inorganic ions, as well as permitting redox activity in the matrix, and facilitating horizontal gene transfer. The retention of water maintains a hydrated microenvironment, protecting against desiccation and proteins and polysaccharides can provide a protective barrier against specific and non-specific host defences during infection, antimicrobial agents and some grazing protozoa (Fig. 3a) (Flemming and Wingender, 2010).

Biofilms and microbial mats have been studied from different perspectives that are relevant to interactions between microorganisms and minerals in a biogeochemical context. Subaerial biofilms occur within solid mineral surfaces exposed to the atmosphere and are dominated by fungi, algae, cyanobacteria and heterotrophic bacteria (Gorbushina, 2007). These communities are known to penetrate the mineral substrates and induce chemical and physical changes contributing to weathering. Effects of biofilms containing the phototrophic cyanobacterium *Nostoc punctiforme* and the rock-inhabiting ascomycete *Knufia petricola* have been quantified using inductively coupled plasma optical emission spectrometry/mass spectrometry as well as scanning electron microscopy/transmission electron microscopy-energy dispersive X-ray spectrometry (Seiffert et al., 2014), demonstrating clear effects of the biofilms on mineral dissolution and leaching. Mats of hypogeous ectomycorrhizal fungi have been studied by Cromack et al. (1979) and Griffiths et al. (1994) found that colonization by the ectomycorrhizal fungus *Gautieria monticola* notably increased the amount of oxalic acid in soil. Calcium oxalate (CaOx) can accumulate in forest soils (Graustein et al., 1977) and deposition of Ca from the weathering of apatite as CaOx crystals on the hyphal surfaces of *Rhizopogon* sp. growing from *Pinus muricata* seedlings has been shown in microcosm studies (Wallander et al., 2002). More CaOx is formed under higher P levels (Tuason et al., 2009). Bulk soil solution concentrations of organic acids are considered to be too low to have a large effect on mineral dissolution, and modelling (Smits, 2009) suggests that local concentrations of weathering agents such as oxalate will not have a major effect on feldspar weathering unless the weathering agents remain within a few microns of the mineral surface. However, several authors (Balogh-Brunstad et al.,





2008; Finlay et al., 2009) have suggested that higher concentrations of organic acids may accumulate within EPS matrices that are in close proximity to mineral surfaces, so that mineral dissolution is influenced, and have called for more experiments to confirm this possible effect. More recent studies by (Gazzè et al., 2013) using atomic force microscopy have demonstrated the presence of EPS halos (Fig. 3b) surrounding *Paxillus involutus* hyphae colonizing phyllosilicate surfaces. In addition to

increasing the surface area for hyphal interaction with mineral surfaces these hydrated EPS layers presumably enhance mineral weathering by promoting accumulation of weathering agents such as organic acids and acidic polysaccharides, but further detailed studies of the local concentrations of these molecules are still necessary.

     Fungi and bacteria live together in a wide range of environments (Deveau et al., 2018) and the exudation of carbon compounds from roots and fungal hyphae into biofilms undoubtedly influences bacterial growth and activity (Guennoc et al.,

2018). Priming of bacterial activity may occur through supply of exudates from vital hyphae (Toljander et al., 2007) but may also include recycling of C from damaged or senescing hyphae (Toljander et al., 2006). Carbon supply from arbuscular mycorrhizal hyphae can provide energy for associated bacteria to solubilize phosphate (Zhang et al., 2014, 2016). Different ectomycorrhizal fungi colonizing lateral roots of tree seedlings have been shown to influence the community structure of associated bacteria (Marupakula et al., 2016, 2017) and differences in the richness and composition of bacterial communities

have been demonstrated between the hyphosphere of ectomycorrhizal fungi and that of saprotrophic fungi (Liu et al., 2018). Although the role of bacteria in mineral weathering has been less widely studied than that of fungi in recent years, progress has been made in understanding the identity and mechanisms of bacteria involved in weathering of minerals in acidic forest soils (Uroz et al., 2009). Bacteria in the genera *Burkholderia* and *Collimonas* appear to have significant mineral weathering ability (Uroz et al., 2011) and burial of different minerals in forest soils appears to result in selection of different bacterial

communities, that are distinct from those of the bulk soil (Uroz et al., 2012), confirming the concept of mineralogical control of fungal and bacterial community structure (Gleeson et al., 2005, 2006; Hutchens, 2010). Uroz et al. (2015) contrasted the rhizosphere with the "mineralosphere" in which bacteria are selected, not by organic nutrients originating from roots, but by the physiochemical properties of different minerals. Recent studies of bacterial assemblages in soil associated with ectomycorrhizal roots of *Pinus massoniana* and *Quercus serrata* have revealed enrichment of oxalotrophic bacteria, degrading

oxalate using the oxalate-carbonate pathway, representing a potential long-term sink for photosynthetically fixed carbon derived from the atmosphere (Sun et al., 2019) (under review).

## 4 Microbial and plant secretions – evidence from microcosms and mesocosms

Plants play a fundamental role in soil formation since root activity and decomposing plant material enhance weathering rates by producing acidifying substances ($H^+$, organic acids), and ligands that complex with metals in the minerals. In addition,

uptake of ions released from weathering reduces the likelihood of saturating conditions that retard weathering rates. Many of these effects are mediated by mycorrhizal fungi and in temperate and boreal forests the vast majority of fine tree roots are colonized by symbiotic ectomycorrhizal fungi.

     In ancient, highly weathered soils, P is the primary nutrient limiting plant growth, whereas N is the main growth-limiting





nutrient in young soils. Plant nutrient acquisition in nutrient-impoverished soils often involves specialized root structures such as cluster roots or symbiotic structures such as mycorrhizas or root nodules (Lambers et al., 2008). In ancient soils with very low P availability 'dauciform' (carrot-shaped) roots are produced by monocots in the Cyperaceae and 'proteoid' roots are produced by numerous dicot families, including the Proteaceae (Fig. 4a-d,). Both types of roots are hairy and produce large

amounts of carboxylates that release P from strongly sorbed forms bound to Al or Fe in acidic soils or Ca in calcareous soils. Phosphatases are also produced to release P from organic sources. Protons are quantitatively important weathering agents and many biotic processes, including uptake of positively charged nutrients such as $NH_4^+$ and $K^+$, result in exudation of protons. Organic acids such as oxalic acid and citric acid are produced by plant roots as well as fungi and bacteria and contribute to proton-driven weathering, but their deprotonated forms also act as strong weathering agents complexing with metals, including

$Fe^{3+}$ and $Al^{3+}$ (Ma et al., 1997, 2001). In acidic environments, organic acid complex formation with $Al^{3+}$ may free P, making it available for plant uptake, whereas in Ca rich environments organic acid exudation by plant roots and subsequent complexation with Ca also increases P availability (Tyler and Ström, 1995). Soil P and N change as a function of soil age and in younger- and intermediate-aged soils with adequate amounts of nutrients, mycorrhizal mycelia provide an effective strategy for nutrient acquisition (Fig. 4e) (Lambers et al., 2008). Experiments using dual isotopic tracers of ($^{14}C$ and $^{33}P$) suggest that

evolution of land plants from rootless gametophytes to rooted sporophytes with larger arbuscular mycorrhizal hyphal networks enabled enhanced efficiency of P capture as atmospheric $CO_2$ concentrations fell during the mid-Palaeozoic (480-360 Ma ago), (Field et al., 2012).

Strategies of mycorrhizal symbiosis differ depending upon the plant host. The majority of plant species form arbuscular mycorrhizas with Glomeromycotan fungi that are efficient at scavenging nutrients such as P, and transporting it to their plant

hosts across the depletion zones around roots formed by the slow diffusion of P through soil. However, these fungi are less efficient than proteoid roots at 'mining' P and releasing it from sorbed forms. Ericoid mycorrhizas are formed by plants in the Ericaceae, Empetraceae and Epacridaceae, and ectomycorrhizas are formed by many woody plants and trees (Smith and Read, 2008). The fungi forming these two types of symbiosis vary in their enzymatic competence but in general they have a more highly developed capacity to both scavenge and 'mine' N and P than arbuscular mycorrhiza, releasing N and P from organic

forms (in the case of ectomycorrhizal fungi) by different combinations of hydrolytic and oxidative enzymes and non-enzymatic Fenton chemistry (Read and Perez-Moreno, 2003; Lindahl and Tunlid, 2015; Nicolás et al., 2018), and P and other mineral elements from inorganic forms via proton, organic acid, and siderophore exudation. In boreal forests with stratified podzol soils, many ectomycorrhizal fungal species produce extensive fungal mycelia that colonize both organic soil horizons and mineral horizons to an equal extent on a land area basis (Söderström, 1979), although data expressed on a soil dry weight basis

often suggest that colonization of the mineral soil is lower since the mineral soil has a dry weight approximately 10 times higher than the organic soil. Studies of vertical distribution of different functional guilds of fungi (Lindahl et al., 2007; Sterkenburg et al., 2018) suggest that ectomycorrhizal fungi are more abundant than saprotrophs in deeper mineral horizons, presumably because they receive supplies of carbon from their plant hosts and are less reliant on local sources of carbon that are less abundant in the deeper mineral horizons.



Mycorrhizal fungal mycelia secrete a wide range of molecules and the secretome has been shown to include low molecular weight (LMW) organic acids, amino acids, polyols, peptides, siderophores, glycoproteins and a diverse range of enzymes such as proteases, phosphatases, lignin peroxidases and laccases. The production of these substances is highly variable both within and between different types of mycorrhizal fungi and influenced by different environmental conditions. Figure 5 illustrates the

flow of plant-derived carbon compounds through the fungal mycelium, the secretion of compounds into extracellular polysaccharide matrices and the soil solution and the longer-term immobilization processes that result in storage of stable C in organic and mineral pools. Although many of the molecules produced by the mycelium and its associated bacteria are labile and subject to rapid turnover, they play a collective role in mobilization of nutrients that can lead to a longer-term sequestration of C in recalcitrant substrates that are both organic (Clemmensen et al., 2013) and inorganic (Sun et al., 2019, under review).

Low molecular weight organic acids are frequently identified as important components of the exudates produced by ectomycorrhizal fungi. Simple carboxylic acids are often present in soil solution and implicated in pedogenic processes. Their sorption characteristics were studied by van Hees et al. (2003) who found adsorbed to solution ratios as high as 3100. Organic acids are readily adsorbed to the solid phase and sorption provides an important buffering role in maintaining soil solution concentrations at low organic acid concentrations, inhibiting microbial degradation of the acids. Concentrations of LMW

organic compounds in soil solution are typically low (<50 μM) but the flux through this pool is extremely rapid and microbial mineralization to $CO_2$ results in mean residence times of 1-10 hours (van Hees et al., 2005). These labile compounds may thus make a substantial contribution to the total efflux of $CO_2$ from soil. Direct measurements of oxalate exudation from hyphal tips of the ectomycorrhizal fungus *Hebeloma crustuliniforme* (van Hees et al., 2006) have led to calculated exudation rates of 19±3 fmol oxalate per hyphal tip per hour, suggesting that concentrations of 30 mM oxalate could occur within one hour inside

feldspar tunnels occupied by fungal hyphae. This would represent a concentration 10 000 times higher than in the surrounding soil solution. In the same study, production of the hydroxamate siderophore ferricrocin was also detected and calculated to be able to reach a concentration of 1.5 μM, around 1000 times higher than in the surrounding soil solution. Interestingly, the steady-state dissolution of goethite by 2'-deoxymugineic acid (DMA) phytosiderophores has been demonstrated to be synergistically enhanced by oxalate (Reichard et al., 2005), and it is possible that synergistic interactions between other

combinations of organic acids and siderophores may exist. Organic acid production by intact ectomycorrhizal fungal mycelia colonizing *Pinus sylvestris* seedlings was studied by Ahonen-Jonnarth et al. (2000), using axenic, *in vitro* systems. In this study, production of oxalic acid (per g root dry weight) by seedlings exposed to elevated (0.7 mM) Al and colonized by *Suillus variegatus* or *Rhizopogon roseolus* was up to 39.5 and 26 times, respectively, higher than in non-mycorrhizal control plants. The same type of lab system was used by Johansson et al. (2008, 2009) to investigate the effect of different mycorrhizal fungi

on production of LMW organic acids, amino acids and DOC. However, in these experiments the identifiable LMW organic acids constituted only a small proportion (3-5%) of the total DOC fraction but DOC production was increased in mycorrhizal treatments relative to the non-mycorrhizal controls.

Studies of mycorrhizal hyphal exudates using NMR spectroscopy (Sun et al., 1999) have revealed exudation of fluid droplets at the hyphal tips of the ectomycorrhizal fungus *Suillus bovinus* and found that sugars and polyols comprised 32%,



and peptides 14% of the exudate mass. Oxalic acids and acetic acid were also found, and polyols such as mannitol and arabitol are thought to be important for retaining turgor in fungal hyphae during C translocation along hydrostatic pressure gradients. High internal pressures in hyphae are thought to be an evolutionary adaptation to facilitate penetration of both plant tissues as well as rock surfaces (Jongmans et al., 1997). This exudation of droplets may play an important role in conditioning the

immediate environment of hyphal tips, facilitating interactions with substrates and associated microorganisms, even in drier soils. Similar observations have been made by Querejeta et al. (2003) who demonstrated that water obtained by *Quercus agrifolia* plants, using hydraulic lift, can be transferred to associated arbuscular mycorrhizal and ectomycorrhizal fungi to maintain their integrity and activity during drought, even when the fertile upper soil is dry. Carbon allocation in the form of sugars and polyols (Sun et al., 1999) may be important in generating turgor pressure in hyphae and have consequences for

weathering of minerals with lattice structure.

While biologically derived molecules such as organic acids and siderophores are strongly implicated in promoting mineral weathering, it is important to note that biologically derived ligands may also inhibit mineral weathering. Among LMW organic acids, only citric and oxalic acids are commonly observed to stimulate mineral weathering (Neaman et al., 2006; Drever and Stillings, 1997), and humic and fulvic acids, which may dominate dissolved organic matter in soil solutions, have been

observed to exert an inhibitory effect on mineral dissolution (Ochs et al., 1996; Drever and Stillings, 1997)

Different microcosm systems have been used to study interactions between minerals and mycorrhizal fungal mycelia colonizing plant seedlings. Differential allocation of plant-derived C to patches of primary minerals such as quartz and potassium feldspar (Rosling et al., 2004) and to apatite and quartz (Smits et al., 2012) suggest tightly coupled plant-fungal interactions underlying weathering. In the experiment by Smits et al. (2012), when P was limiting, 17 times more $^{14}$C was

allocated to wells containing apatite than to those containing only quartz, and fungal colonization of the substrate increased the release of P by a factor of almost three (Fig. 6). Experiments by van Schöll et al. (2006a) demonstrated that limitation of nutrients (P, Mg, K) affected the composition of organic acids exuded by ectomycorrhizal fungi (more oxalate) but not the total amounts. Other experiments by van Schöll et al. (2006b) have demonstrated significant weathering of muscovite by the ectomycorrhizal fungus *Paxillus involutus* when K was in low supply whereas no effect on hornblende was found under Mg

deficiency. Selective allocation of biomass to grains of different minerals by *P. involutus* has also been demonstrated (Leake et al., 2008; Smits et al., 2008) suggesting grain scale "biosensing", however it is also possible that fungal growth may be influenced by topographic structure (Smits and Wallander, 2017). Schmalenberger et al. (2015) demonstrated mineral-specific exudation of oxalate by *P. involutus* using labelled $^{14}$CO$_2$ given to the host plant. Oxalate was exuded in response to minerals in the following sequence: Gabbro > limestone, olivine and basalt > granite and quartz. Experiments using flow-through

systems (Calvaruso et al., 2013) have also estimated weathering rates of apatite to be 10 times higher when pine seedlings were present, compared with unplanted systems and attributed this to exudation of organic acids by the roots. The plants had been checked for the absence of fungal 'contaminants' but inoculation with the mineral weathering bacterial strain *Burkholderia glathei* PML1(12)Rp appeared to have no significant effect on weathering.



Fungi, bacteria and plants all produce siderophores, low-molecular-mass, metal-complexing compounds. These bind strongly to $Fe^{3+}$ playing key roles in its release and uptake (Kraemer et al., 2014; Ahmed and Holmström, 2014). The hydroxamate siderophores ferrichrome and ferricrocin have been found in soil solution of mor layer podzolic soil overlying granitic rock and intensively colonized by ectomycorrhizal hyphae (Holmström et al., 2004) and should be kinetically and

thermodynamically, even more efficient complexing agents for trivalent cations than oxalic and citric acid. Primary minerals containing substantial amounts of Fe, such as hornblende and biotite, show enhanced dissolution rates in the presence of microbial or fungal siderophores (Kalinowski et al., 2000; Sokolova et al., 2010) and attachment of microorganisms to the mineral surfaces appears to lead to greater dissolution of elements from biotite (Bonneville et al., 2009; Ahmed and Holmström, 2015)

Release of potassium from K-feldspar and illite in microcosms by the fungus *Aspergillus fumigatus* was demonstrated by Lian et al. (2008) who showed that release of K was enhanced by a factor of 3-4 by physical contact between the fungus and the mineral surface. Simple types of microcosm are usually used for gene expression studies in order to facilitate extraction of RNA from target organisms. Xiao et al. (2012) used differential expression cDNA libraries of *A. fumigatus* using suppression subtractive hybridization (SSH) technology to investigate the mechanisms by which the fungus weathered K-bearing minerals.

K-bearing minerals were found to upregulate the expression of carbonic anhydrase (CA), implying that *A. fumigatus* was capable of converting $CO_2$ into carbonate to accelerate the weathering of potassium-bearing minerals, which fixed $CO_2$. During mineral weathering, the fungus changed its metabolism, produced more metal-binding proteins, and reduced membrane metal transporter expression, which can modulate ion absorption and disposal and promote acid production. Wang et al. (2015) used high-throughput RNA-sequencing (RNA-seq) to study the molecular mechanisms of *Aspergillus niger* involved in weathering

of potassium feldspar. The fungus was cultured with soluble $K^+$ or K-feldspar) demonstrating differential expression of genes related to synthesis and transportation of organic acids, polysaccharides and proteins which was closely related to release of $K^+$ from the minerals. Regulation of carbonic anhydrase (CA) gene expression in *Bacillus mucilaginosus* and the effects of its expression product in *Escherichia coli* have been examined by Xiao et al. (2014) who found that expression of CA genes was upregulated by the addition of calcite to a $Ca^{2+}$-deficient medium, and that a crude enzyme extract of the expression product

in *E. coli* promoted calcite dissolution. Real-time fluorescent, quantitative PCR has been used to explore the correlation between CA gene expression in *B. mucilaginosus* and deficiency/sufficiency of Ca and $CO_2$ concentration (Xiao et al., 2015) and the results suggest that CA gene expression is negatively correlated with both $CO_2$ concentration and the ease of obtaining soluble calcium (Xiao et al., 2015). The roles of different CA genes have also been studied in *Aspergillus nidulans* using gene deletion, overexpression and bioinformatics (Sun et al., 2019) and the results of this study suggest that the CA gene *canA* is

involved in weathering of silicate minerals and carbonate formation, catalysing $CO_2$ hydration and that *canB* is essential for cellular respiration and biosynthesis in low $CO_2$ environments.





## 5 Systemic consequences of microorganism-mineral interactions in an ecological and evolutionary context

There is strong support for the idea that microorganism-mineral interactions have important systemic consequences at global spatial scales and evolutionary time scales. Indeed, the concept of "mineral evolution" (Hazen et al., 2008) suggests that *over two thirds of the number of minerals that exist today (>5300) are the result of chemical changes mediated by living organisms*.

The best known of these is the Great Oxidation Event about 2.3 billion years ago (2.3 Ga) (Kump, 2008; Luo et al., 2016) during which the Earth's atmosphere changed from one that was almost devoid of oxygen to one that is one-fifth oxygen. The discovery of inclusions of potentially biogenic carbon (with a high $^{12}C/^{13}C$ ratio) within Hadean zircons as old as 4.1 Ga (Bell et al., 2015) suggests that biological processes could have been operating during the Hadean Eon. Early microbial communities would have developed within sub-surface mineral environments to avoid high levels of ionizing radiation at the interface

between the atmosphere and lithosphere. The sub-aerial biofilms that exist at this interface today remain stressful environments (Gorbushina, 2007) but ionizing radiation levels are now much lower due to the thickening of the Earth's atmosphere, thought to be caused by the increase of the magnetic field dipole intensity due to the solidification of the inner core, caused by the cooling of the Earth (Doglioni et al., 2016). Biomarker evidence (Brocks et al., 1999) in rocks that formed 200 million years (Ma) before the increase in atmospheric oxygen suggests that oxygen was already being produced before 2.5 Ga. Oxygenic

photosynthesis by cyanobacteria is a likely source of this oxygen but there is evidence that stromatolites were abundant between 3.4 and 2.4 Ga, prior to the advent of cyanobacteria and oxygenic photosynthesis (Allen, 2016) and that Archaean microbial mats of protocyanobacteria switched between photolithoautotrophic and photoorganoheterotrophic metabolism prior to the evolution of cyanobacteria with simultaneous constitutive expression of genes allowing both types of metabolism. It is also likely that phototrophy based on purple retinal pigments similar to the chromoprotein bacteriorhodopsin, discovered in

halophilic Archaea, may have dominated prior to the development of photosynthesis (DasSarma and Schweiterman, 2018). The activity of these early microorganisms and the subsequent accumulation of oxygen in the atmosphere paved the way for the evolution of plants and there is a large and diverse body of evidence that the plastids of algae and higher plants evolved from free-living bacteria by endosymbiosis involving endosymbiotic gene transfer (Zimorski et al., 2014) as well as horizontal gene transfer (Archibald, 2015).

  The evolution of higher plants and development of vegetation has had a substantial effect on soil mineral weathering (Berner, 1992; Berner et al., 1983). The first well differentiated forests appeared in the Devonian, and increases in the volume of roots from the Silurian to the Devonian are associated with increases in clay enrichment chemical weathering in subsurface horizons and draw down of atmospheric $CO_2$ (Retallack, 1997). Dissolution of bed rocks, accelerated by the growth of plants

and enhanced weathering of silicates, resulting in $HCO_3^-$ carried to the sea and precipitated as carbonates, would have led to removal of $CO_2$ from the atmosphere and the large drop in $CO_2$ during the Devonian 400-360 Ma ago is thought to be associated with the rise of land plants and spread and development of forests (Berner, 1997).



The ubiquitous distribution of microorganisms today supports the idea that plants (and all other higher organisms) are not standalone entities; rather, they should be considered from a more holistic perspective, as **holobionts**, including the full diversity of the many different microorganisms associated with them (Vandenkoornhuyse et al., 2015). The fact that almost all plant roots are colonized by microbial symbionts makes it difficult to quantify the separate contributions of plants and

associated microorganisms to mineral weathering. There is broad agreement that fungi are important biotic agents of geochemical change (see Gadd, 2007, 2010, 2013 a, b, 2017) and that the coevolution of fungi and plants has enabled them to have increasing influence as biogeochemical engineers. Fungi exert a significant influence on biogeochemical processes, especially in soil, rock and mineral surfaces and the plant root-soil interface, where, as mycorrhizal fungi, they are responsible for major mineral transformations, redistribution of inorganic nutrients and flow of C. They are important components in rock

inhabiting communities with roles in mineral dissolution and secondary mineral formation. The ubiquity and significance of lichens as pioneer organisms in the early stages of mineral soil formation, and as a model for understanding weathering in a wider context, are well understood (Banfield et al., 1999; Chizhikova et al., 2016).

However, many studies of ectomycorrhizal influence on weathering rates have been performed over short periods under laboratory conditions and there is often no clear evidence that processes observed at the laboratory-scale play a significant role

in "soil-scale" mineral dissolution rates. Evidence from field experiments does not always provide unequivocal evidence for a major role in weathering, either. As an attempt to span a longer time-scale for biological weathering studies, Smits and Wallander (2017) used a vegetation gradient from bare soil, via sparse grass to Norway spruce forest in a natural lead contaminated area in Norway. This gradient had probably been present since the last glaciation and made it possible to study long-term effects of vegetation on apatite weathering in moraine deposited at the end of the last glaciation. The presence of

vegetation had a strong stimulatory effect on apatite weathering. Mainly because of the acidifying effect of plant growth. In fact, 75% of the variation in apatite weathering could be explained by soil pH. The effect of plant roots and mycorrhizal symbionts on this process could not be separated. However, in the top 20 cm of the mineral soil an additional mechanism not mediated by pH, was active that furthered enhanced dissolution of apatite (Fig. 7). The authors suggested this to be LMW organic acids which may contribute to apatite weathering especially at higher pH when they are deprotonated, leading to higher

concentrations of the organic-metal complexes on the mineral surfaces. However, the origin of these acids is probably not ectomycorrhizal fungi in the study by Smits et al. (2014) since these fungi were absent in the grass vegetation at the highest pH of the vegetation gradient. Under these conditions, the biomechanical and chemical effects of ectomycorrhizal fungi on apatite weathering seemed to be minor, but these effects are probably dependent on the nutrient status of the forest. Enhanced colonization of apatite by ectomycorrhizal hyphae in laboratory systems (Rosling et al., 2004; Smits et al., 2012) is also

commonly found under field conditions, but only when P availability is low (Rosenstock et al., 2016; Bahr et al., 2015; Almeida et al., 2018). The potential for weathering by ectomycorrhizal fungi is probably much higher under these conditions and the nutrient status of the forest should be considered when biological weathering rates are quantified, at least for apatite weathering, where P status has a strong effect on fungal colonization of apatite. In contrast, no enhanced colonization of biotite and hornblende by ectomycorrhizal hyphae was found in Norway spruce forests in the Czech Republic under low K or Mg





availability (Rosenstock et al., 2016). This suggests that ectomycorrhizal fungi have a smaller potential to enhance weathering of these minerals compared to apatite. However, these results should be treated with caution since no quantitative/chemical estimates of the mineral weathering were made in this study and use of ergosterol based increase or decrease in fungal biomass as a proxy for 'weathering' can be misrepresentative, since some ectomycorrhizal fungi that actively release LMW organic

acids may not invest much carbon in their own biomass. Further investigations using RNA-based analysis of active fungal and bacterial communities, combined with temporal assessment of weathering kinetics, should reveal the true potential of microorganisms in biogeochemical weathering in forest ecosystems.

### 5.1 Weathering and carbon allocation and sequestration

The role of fungi in weathering of rocks and minerals through biomechanical and biochemical attack has been extensively

studied. Proton-promoted dissolution is supplemented by ligand-promoted dissolution of minerals by strong chelators such as oxalic and citric acid that may act synergistically with siderophores. Secondary minerals may be deposited as carbonates, oxalates or other mycogenic minerals and mineraloids and the role of "rock-building fungi" has been discussed in addition to the role of "rock-eating fungi" (Fomina et al., 2010). Fungi, including many ectomycorrhizal fungi, are prolific producers of oxalate, and oxalotrophic bacteria are capable of oxidizing calcium oxalate to calcium carbonate. Since the oxalate is organic

in origin, and half of its C is transformed into mineral C with a much longer residence time than organic C, this process represents a potential major sink for sequestration of C from the atmosphere (Verrecchia et al., 2006). Precipitation of carbonate minerals by microorganisms in connection with silicate weathering has also been discussed by Ferris et al. (1994) in relation to its potential role as a sink for atmospheric carbon dioxide. The oxalate-carbonate pathway may not be important in boreal forest soils, however the African oxalogenic iroko tree *Milicia excelsa,* together with associated saprotrophic fungi

and bacteria, enhances carbonate precipitation in tropical oxisols, where such accumulations are not expected due to the acidic nature of the soil (Cailleau et al., 2011). The same phenomenon has been demonstrated in acidic soils of a Bolivian tropical forest (Cailleau et al., 2014) although the significance of the pathway to the global carbon balance of the tropics has not been demonstrated. The role of microorganisms in dissolution and modification of karst stones such as limestone and dolomite has also been studied (Lian et al., 2010, 2011). Microbially mediated chemical corrosion and precipitation in surface and

underground water can play a role in pedogenesis and provide a sink for atmospheric $CO_2$ and the role of carbonic anhydrase in hydrating atmospheric $CO_2$ to $HCO_3^-$ has been investigated in relation to changes in $CO_2$ concentration and availability of $Ca^{2+}$ (Xiao et. al., 2014, 2015). The results of the latter study suggest that the importance of microbial carbonic anhydrase on silicate weathering and carbonate formation may be higher at current $CO_2$ levels than under primordial conditions 2 Ma ago when $CO_2$ levels were much higher.

Experiments by Högberg et al. (2001), demonstrating rapid decreases in soil respiration following the girdling of forest trees, suggest that the flux of current assimilates to mycorrhizal roots is directly connected to the supply and respiration of C in soil. In another study (Högberg and Högberg, 2002), extractable DOC in a 50-year-old boreal forest, was 45% lower in girdled plots than in control plots, suggesting a large contribution by roots and associated fungi to soluble C pools, although




the contribution of these two components could not be determined separately. In 15-year-old pine stands, the mean residence time of labile C in ectomycorrhizal root tips has been estimated to be about 4 days, and that of total soil microbial cytoplasmic C to be 17 days in field experiments using $^{13}CO_2$ tracer (Högberg et al., 2008), suggesting rapid carbon through-flow in fungal biomass. Similar rapid transport of C through arbuscular-mycorrhizal mycelial systems has been shown in soil from upland

grassland ecosystems by Johnson et al. (2002a, b), although this flow through the fungal networks may be disrupted by the activity of grazing soil arthropods (Johnson et al., 2005).

Biogeochemical weathering of silicate rocks is a key process in the carbon cycle (Pagini et al., 2009) and, although consumption of $CO_2$ by weathering is small compared with transfers associated with photosynthesis and respiration, it is the dominant sink in the global carbon balance and controls atmospheric $CO_2$ and climate patterns at scales of millennia or longer

(Goudie and Viles, 2012). Catchment-scale field studies consistently indicate that vegetation increases silicate rock weathering but incorporating the effects of trees and fungal symbionts into geochemical carbon cycle models has relied upon simple empirical scaling functions. Taylor et al. (2012) developed and applied a process-based approach to derive quantitative estimates of weathering by plant roots, associated symbiotic mycorrhizal fungi and climate, concluding that vegetation and mycorrhizal fungi enhance climate-driven weathering by a factor of up to two.

The geoengineering potential of artificially enhanced silicate weathering is now increasingly well established (Köhler et al., 2010) and addition of pulverised silicate rocks to different croplands has now been advocated as an effective strategy for global carbon dioxide removal (CDR) and ameliorating ocean acidification by 2100 (Taylor et al., 2016; Beerling et al., 2018). Large scale field trials are now in progress http://lc3m.org/ but basic information about the way in which different microorganisms drive the sequestration processes in different soil types is still missing. Recent studies on carbonate weathering

by ectomycorrhizal fungi colonizing tree roots (Thorley et al., 2015) suggest that ectomycorrhizal tree species weather calcite containing rock grains more rapidly than arbuscular mycorrhizal (AM) trees because of greater acidification by the ectomycorrhizal trees. Weathering and corresponding alkalinity export to oceans may increase with rising atmospheric $CO_2$ (Andrews and Schlesinger, 2001) and associated climate change, slowing rates of ocean acidification.

Successive increases in the size of plant hosts and the extent of substrate colonization by their fungal symbionts (Quirk et

al., 2015) have enabled them to have larger effects as biogeochemical engineers, affecting the cycling of nutrients and C at an ecosystem and global level. It is accepted that ectomycorrhizal fungi access and degrade organic nitrogen sources and it has been shown that soil carbon storage is greater in ecosystems dominated by ectomycorrhizal plants than in systems dominated by other types of mycorrhiza (Averill et al., 2014). Transfer of increasing amounts of photosynthetically derived carbon to ectomycorrhizal fungi and improved colonization of mineral substrates during evolution of plants (Quirk et al., 2012, 2014)

are consistent with the idea that weathering of silicate minerals and sequestration of C into ocean carbonates has led to draw down of global $CO_2$ levels during the rise of ectomycorrhizal trees over the past 120 Ma (Taylor et al., 2009, 2011; Morris et al., 2015). However the relative constancy of atmospheric $CO_2$ levels and absence of even further reductions over the final 24 Ma of the Cenozoic has been attributed to a negative feedback mechanism caused by $CO_2$ starvation (Beerling et al., 2012)



that is predicted, by numerical simulations, to reduce the capacity of the terrestrial biosphere to weather silicate rocks by a factor of four.

Inferences about the evolutionary development of weathering have been drawn by using vertical scanning interferometry to study "trenching" of silicate mineral surfaces (basalt) buried under different tree species growing in an arboretum (Quirk et

al., 2012). These studies suggest that both trenching and hyphal colonization increased with evolutionary progression from AM fungi to ectomycorrhizal fungi, and with progression from gymnosperm to angiosperm host plants. It is suggested that this evolutionary progression resulted in a release of calcium from the basalt by ectomycorrhizal gymnosperms and angiosperms at twice the rate achieved by AM gymnosperms, and that forested ecosystems have become major engines of continental silicate weathering, regulating global $CO_2$ concentrations by driving calcium export into ocean carbonates (Quirk

et al., 2012) (Fig. 8). Additional laboratory studies of the same tree species using different $CO_2$ environments suggest that weathering intensified during evolutionary progression from AM fungal symbionts to ectomycorrhizal symbionts and that calcium dissolution rates were related to photosynthate energy fluxes and were higher during simulated past $CO_2$ atmosphere (1500 ppm) under which ectomycorrhizal fungi evolved (Quirk et al., 2014).

### 6 Methods using stable isotopes

Stable isotopes, especially of Ca and Sr, have been used extensively to source the origin of Ca in drainage water; when applied to plant tissues, they can be used to trace plant nutrients back to their primary source. Isotope tracing has been mostly used to study apatite weathering. Apatite is a calcium–phosphate mineral, and because P has no stable isotopes, the uptake dynamics can only be studied via the Ca ion (or potentially the $^{18}O/^{16}O$ in the phosphate group). In most rocks and soils, apatite is the sole primary P source. However, its contribution to the soil solution Ca pool is minor compared with other minerals. If the Ca

isotope ratio in the plant is more similar to the signature in apatite than to the signature in the soil solution, then it indicates that the plant directly acquires Ca from apatite. Blum et al. (2002) applied this technique to a temperate mixed forest using Ca:Sr ratios in soil water, minerals in the soil and different mycorrhizal and non-mycorrhizal trees. The authors concluded that direct calcium uptake by ectomycorrhizal fungi weathering apatite in the parental material could compensate for calcium loss in base-poor ecosystems. Data on element ratios should, however, be interpreted with care, because of high variation of Ca:Sr

ratios in different plant tissues and limited understanding of the cycling of these elements in plants (Watmough and Dillon, 2003). Field studies using mesh bags containing microcline and biotite, buried in Swedish *Picea abies* forests (Wallander et al., 2006) used the $^{87}Sr:^{86}Sr$ ratio to calculate the fraction of Sr in the mycorrhizal root tips that had originated from the minerals. Although the total amounts of Sr released from the minerals could not be calculated since the total plant biomass enriched with $^{87}Sr$ was unknown, the study clearly demonstrates the potential of ectomycorrhizal fungi to mobilize and take up nutrients such

as Ca and K from microcline and biotite under field conditions.

In many forest ecosystems, plant-available pools of Mg, Ca, and K are assumed to be stored in the soil as exchangeable cations adsorbed on the cation exchange complex (exchangeable pools). However, other storage forms of Mg, Ca, and K that





have not been fully characterized may play an important role in plant nutrition and biogeochemical cycles and be plant-available on very short time scales (<1 day). Isotopic dilution techniques using the stable isotopes $^{26}$Mg, $^{44}$Ca, and $^{41}$K have been developed (van der Heijden et al., 2018) to trace and quantify the pools of Mg, Ca, and K (isotopically exchangeable pools) in the soil of a hardwood forest that contribute directly to equilibrium processes between the soil water and the soil.

These show that isotopically exchangeable pools of Mg, Ca, and K are greater than traditionally measured exchangeable pools. Storage forms of Mg, Ca, and K in the isotopically exchangeable pool could include chelation with soil organic matter, retention on soil aluminum and iron oxides and hydroxides through phosphate and/or organic acid bridges and site-specific adsorption. The isotopic dilution method is a relevant tool to quantify the plant-available pools of Mg, Ca, and K on short time scales (source and sink pools) and is a very promising approach to characterize and quantify the processes responsible for the

depletion and/or replenishment of these pools over longer time scales.

Field studies of small rock fragments isolated from a Finnish *P. sylvestris* forest with *Tricholoma matsutake* fruiting bodies (Vaario et al., 2015) revealed the presence of *T. matsutake* on 97% of the rock fragments and laboratory assays using X-ray diffraction confirmed the ability of the fungus to absorb some trace elements directly from the rock fragments, but uptake of Mg and K did not appear to be significant. In contrast, laboratory studies of the capacity of different fungi to mobilize P and

base cations from granite particles (conducted within QWARTS) (Fahad et al., 2016) suggest that some ectomycorrhizal fungi can mobilize and accumulate significantly higher concentrations of Mg, K and P than non-mycorrhizal fungi. The mycorrhizal fungi can fractionate Mg isotopes, discriminating against heavier isotopes and we found a highly significant inverse relationship between $\delta^{26}$Mg tissue signatures and mycelial concentration of Mg (Fig. 9). This provides a theoretical framework for testing hypotheses about fungal weathering of minerals in future experiments. If active mobilization and uptake of lighter

$^{24}$Mg isotopes results in relative enrichment of heavy Mg isotopes left in soil solution and soil, this should be evident in areas of active weathering. Mesocosm experiments, conducted within the QWARTS project (Mahmood et al., in preparation), employing a gradient of increasing organic matter depletion to simulate progressively more intense forest biomass harvesting, revealed significant and successive enrichment of $^{26}$Mg signatures in the soil solution in the B horizon, associated with increased availability of organic matter and resultant increases in plant and fungal biomass (Fig. 10). No such enrichment was

found in other horizons or in systems without plants (and therefore without mycorrhizal fungi). This suggests that significant biological weathering of Mg takes place in the B horizon, driven by higher plant biomass that enables improved carbon allocation to the fungal mycelium and also constitutes a larger sink for uptake of mobilized base cations. Although the experiments provide strong support for the idea of biologically driven mobilization of Mg from B horizon mineral soil, the process was not sufficient to maintain optimal tree growth in systems with a severely reduced organic matter pool. In addition,

studies carried out under both field and laboratory conditions show that Mg isotope fractionations are controlled by the same biological factors in the critical zone, defined as the outer layer of earth from vegetation to the soil. Silicate rocks show a relatively small range of variation in Mg isotopic ratios (denoted as $\delta^{26}$Mg) (Tipper et al., 2006; Brenot et al., 2008; Bolou-Bi et al., 2009; Shen et al., 2009; Uhlig, et al., 2017). During the weathering of these rocks at watershed level, it was revealed that isotopic fractionation of Mg isotopes was in favour of light isotopes in soil solution, while the soils were enriched in heavy



isotopes (Tipper et al., 2006; Brenot et al., 2008; Pogge von Strandmann et al., 2008; Tipper et al., 2010). Studies conducted in forest ecosystems, (Bolou-Bi et al., 2012; Mavromatis et al., 2014; Uhlig et al., 2017) indicate variation in soil solution signatures of surface soil layers, suggesting a role of vegetation through the Mg isotope cycle (uptake and litterfall), soil exchangeable fraction and rainwater, in addition to light Mg isotope return via litterfall. In deeper soil horizons, however, the

soil solution signatures may be the result of two additional processes (a) the mineral dissolution leaching the light isotope into solution and subsequently weathered minerals are systematically enriched in heavy Mg isotopes relative to fresh rock, and (b) clay formation and/or Mg adsorption removing the heavy Mg isotope from soil solution (Huang et al., 2012; Opfergelt et al., 2014). Mg isotope fractionation has also been observed under laboratory conditions during the dissolution of primary minerals (Wimpenny et al., 2010).

In studies of Ca isotope cycling in forest ecosystem, it appears that the soil solution and exchangeable fraction generally display enrichment in the heavy isotope compared to soil particles, bedrock and rainwater (Cenki-Tok et al., 2009; Holmden and Bélanger, 2010; Hindshaw et al., 2011). However, the soil solution isotope signatures are not the simple result of weathering processes in soils because the congruent dissolution of rock or mineral observed in lab and field conditions did not cause any measurable Ca isotope fractionation (Hindshaw et al., 2011; Tipper et al., 2006; Ryu et al., 2011; Cobert et al.,

2011). This suggests that another process, such as the preferential uptake of the light Ca isotope ($^{40}$Ca) by vegetation, decreases the soil solution Ca isotope ratio in the upper horizon in addition to light Ca isotope return via litterfall (Page et al., 2008; Cenki-Tok et al., 2009; Holmden and Bélanger, 2010; Cobert et al., 2012). In deeper soil horizons, soil solution $\delta^{44/40}$Ca may result from the dissolution of minerals such as apatite. Interestingly, experiments by Dijkstra and Smits (2002) indicate that most of the Ca taken up by trees comes from litter recycling. In a comparable mixed forest, also in the north-eastern United

States, the annual Ca import from weathering in the rooting zone is less than 0.3% of the annual Ca uptake, which was a four-fold smaller flux than the annual atmospheric deposition. The data from our QWARTS experiments suggest that mobilization of Mg may function differently with higher amounts being mobilized from inorganic substrates in the B horizon.

Mycorrhizal fungi play a central role in mobilizing N and P from organic substrates and when these are depleted, N and P limit tree growth, resulting in reduced C supply to the mycorrhizal mycelium and reduced capacity for mobilization of base

cations from the mineral horizons. Although mobilization of Mg from the B horizon was sufficient to support increased biomass production in systems supplied with extra organic material (Fig. 10), it was not sufficient to compensate for losses of base cations when organic material was most depleted. The results of these experiments are therefore consistent with the predictions of modelling that, under intensive forestry with removal of organic residues, base cation supply will not be sustainable in the long term. Intensive, sustained harvesting of biomass may lead to N limitation before base cations become

limiting. Applications of different fertilizers (Xiao et al., 2017) or inadvertent N deposition (Averill et al., 2018) may have negative effects on both weathering and C sequestration.

Despite the large number of laboratory experiments (e.g. Smits et al., 2012) demonstrating that ectomycorrhizal fungi increase weathering of minerals Smits and Wallander (2017) consider that there is no clear evidence that processes observed



at the laboratory-scale play a significant role in "soil-scale" mineral dissolution rates. Detailed studies of the liquid chemistry of local weathering sites at the micrometre scale, together with up-scaling to soil-scale dissolution rates, are advocated and the authors suggest that future research should focus on whole ecosystem dynamics, including the behaviour of soil organic matter, and that early-stage primary succession ecosystems on low reactive surfaces, such as fresh granites, should be included. Smits

and Wallander (2017) also recommend the use of stable isotopes by choosing minerals and soils with distinct isotope ratios.

## 7 Modelling of weathering in forest soils

Akselsson et al. (2019) present an extensive review of methods for estimating weathering rates in forest soils. These range from mass balance budget calculations (e.g. Simonsson et al., 2015), to the depletion method based on the elemental concentration differences between of weatherable and unweatherable (such as zirconium) elements, to dynamic models based

on the transition state theory (e.g. Stendhal et al., 2013; Erlandsson et al., 2016). One such model is implemented in the PROFILE and ForSAFE models which have been widely used for unsaturated soils (for recent examples see Akselsson et al., 2016; Belyazid et al., 2019; Phelan et al., 2014).

Mineral weathering is currently fully dynamically simulated in the PROFILE and ForSAFE models (Erlandsson et al., 2016; Belyazid et al., 2019). In ForSAFE, tree cover, soil microbes and related biological processes are also integrally simulated. All

processes described here refer to how they are simulated in the model based on best available empirical and theoretical knowledge. Trees are assumed to affect weathering through a number of causal pathways: Firstly, trees have a direct negative influence on soil moisture through transpiration. The consequent reduction in soil moisture limits weathering directly, as the latter is directly dependent on water, as one actor responsible for the dissolution of the mineral matrices and the subsequent release of elements. Water uptake also leads to an increase in element concentrations, which in turn activates the so called

weathering brakes that slow down the weathering rates as solute concentrations increase according to the principles of the transition state theory governing the weathering equations (Erlandsson et al., 2016). Secondly, nutrient uptake reduces the concentration of weathering products (potassium, magnesium and calcium in particular), releasing these brakes, thereby promoting weathering rates. Thirdly, plants are responsible for the production of organic matter, which, through below ground allocation and litter fall, feeds soil organic carbon, dissolved organic carbon and $CO_2$ concentrations. Both organic radicals

and higher soil $CO_2$ pressure have positive influences on weathering rates. At the same time, the decomposition of litter also releases the base cations contained in the former, thus increasing element concentrations and thereby slowing down weathering rates. The role of organic radicals and $CO_2$ pressure overrides the inhibition from the release of base cations. Fourthly, plants have a direct effect on soil solution pH which in turn drives one of the four simulated weathering pathways. According to the kinetic response assumed in the model, lower pH promotes higher mineral dissolution, i.e. weathering. Plants can lower pH

through the production of organic matter, but most importantly through the uptake of positively charged cations. To preserve the charge balance in the soil solution as cations are removed through uptake, the model assumes that protons are mobilized accordingly, thus lowering soil solution pH. Lower pH in turn promotes higher weathering rates, but also the solubility of





aluminium ions whose higher concentrations act as weathering brakes. The net effect on weathering, i.e. the balance between the positive effect from lower pH and the negative from higher aluminium concentrations, may differ depending on soil properties. ForSAFE does not treat the belowground physiology of trees as a function of mycorrhizal type, but, as each tree species has a discreet parameter set, including root distribution and activity, mycorrhizal weathering promoting activities can

be implemented largely within existing process descriptions. The contribution of biological weathering may be improved by division of the soil volume into rhizosphere or mycorrhizosphere and bulk soil portions and improved process descriptions of root and hyphal influence on the solution and surface chemistry within the rhizosphere/mycorrhizosphere. Division of DOC into discrete chemical functional classes (promoting vs inhibitory, actively exuded vs. incomplete decomposition products) could increase our understanding of the influence of ligand promotion or inhibition of weathering rates.

The interplay between vegetation dynamics and weathering rates is said to happen in close proximity to root tips, where solution chemistry at microscopic level may differ greatly from that of the bulk soil solution. The contact area between hyphae and mineral surfaces is increased by EPS haloes (Gazzè et al., 2013) and many fungal exudation products such as organic acids and siderophores may be released into polysaccharide matrices in close proximity to mineral surfaces. This distinction is not considered in the models, and may therefore cause the simulations to overlook feedback mechanisms by which trees regulate

the concentrations of active molecules to promote weathering, or immobilize toxic elements and thereby alleviate the negative effect of solution saturation on weathering rates. Another process currently overlooked in the models but that may need further attention is the active exudation of carbon compounds to stimulate weathering. The drivers of exudation are currently not well understood, making the simulation of the entire process and its eventual effects on weathering uncertain.

**7.1 Information needs and possible improvements**

For models to be useful as predictive tools, to understand future responses to land management or global change, they must be mechanistic, and describe explicitly processes that significantly impact the desired model outputs. At the same time, each additional layer of complexity, additional parameter and equation, and value that needs to be calibrated makes a model more difficult to apply across a range of settings, and may reduce the usefulness of the model. The appropriate level of complexity and the right set of processes and parameters for models that seek to quantify soil mineral weathering rates will ultimately

depend on the particular time and spatial scale of the enquiry, as well as the quality of data available, but there is ample evidence that, today, we lack models that are sufficiently mechanistic with regard to biological weathering to allow us to determine, with sufficient certainty, what levels of extraction from forests are sustainable, sustainable (acidic) deposition levels on catchments, or how patterns of global primary productivity may change over the next century in response to global change.

    In the preceding sections we have discussed the importance of microbial exudates, biofilms, and mechanical forcing in

biological weathering, as well as the dependence of these processes on plant and mycorrhizal functional groups and belowground carbon allocation by plants. We have identified five processes that can be incorporated into models quantifying soil mineral weathering rates, to make them more mechanistic and useful as predictive tools:



- Exudation of ligands promoting weathering
- Nutrient uptake rates as a driver of weathering reactions
- The concentration effect of biofilms on weathering-promoting ligands and protons
- The dependence of the above process on particular microbial and plant assemblages, including mycorrhizal type.
- The dependence of the above processes on carbon flux from autotrophs, and the sensitivity of this to water and nutrient availability.

The aforementioned stimulatory effect of particular LMW organic acids and siderophores on soil mineral weathering rates is a function of both exudation rate as well as biological degradation rates of the same compounds. The chemical makeup of DOC and exudates exerts considerable control on the degree of stimulation, or, potentially, inhibition of weathering rates, and

modelling ligand-promoted dissolution as a function of DOC does not reflect this and may lead to inaccurate interpretations of mineral dissolution rates derived from experiments with specific organic species. Geochemical weathering models require the incorporation of mechanisms (either through equilibrium equations or inhibitory factors or "brakes") that allow the build-up of weathering products to slow weathering rates, as discussed below. The treatment of soil solution fluxes should allow for accumulation of weathering products in soil microenvironments such as around mineral surfaces and mycelia, as evidence

suggests that bulk soil solution data may not capture these concentration gradients found around mineral surfaces. As noted above, the stimulatory effects of exudates are highly concentration dependent, and many studies fail to find sufficiently high concentrations of weathering promoting ligands in bulk solution, while other studies indicate that biofilms may allow these compounds to concentrate by orders of magnitude near mineral surfaces, increasing their effectiveness as weathering agents. Similarly, precipitation accelerating dissolution, bio-acidification and $CO_2$ respiration effects are likely to be magnified under

biofilms.

The dependence of the above processes on particular microbial and plant assemblages, in particular mycorrhizal type, has not yet been determined and requires further detailed studies of the specificity of ectomycorrhiza-bacteria interactions and the physiological differences between different bacteria and fungi in their ability to promote dissolution of different minerals Further information is also required on the variation in carbon flux from autotrophs to different fungal and bacterial

components of biofilms in contact with different minerals and the sensitivity of this carbon flux to differences in water and nutrient availability.

Catchment and ecosystem mass balance approaches have been widely employed to estimate mineral weathering rates. The simplest among them assume that the soil and standing biomass nutrient pools are at steady state, and mineral weathering rates are equivalent to the difference between stream outflows and atmospheric deposition. More complex approaches have

attempted to account for changes in soil and biomass pools, as well as to measure leaching, but a general feature of mass-balance approaches is that weathering rates are derived from the sum of inputs and outputs into other pools in the system. Models have been developed to describe increasingly complex sets of processes, as well as to apportion calculated weathering fluxes into individual contributing mineral species (Price et al., 2013). Meta-analysis across large numbers of studies



(Hartmann and Moosdorf, 2011) and the application of machine learning approaches (Povak et al., 2014) can further increase the utility of mass-balance approaches for evaluating the potential importance of different processes to weathering rates. Biomass accumulation emerges in many simulations (Wilcke et al., 2017; Zetterberg et al., 2016) as a key flux controlling the dissolution and retention rates of mineral weathering products and a major source of uncertainty (Simonsson et al., 2015;

Zetterberg et al., 2016). The uncertainty of calculated weathering rates is often quite large in comparison to the rate of weathering, as a result of the magnitude of the weathering flux often being far smaller than the fluxes from which they are calculated (Simonsson et al., 2015). While mass-balance approaches, given careful scrutiny of inherent assumptions, are valuable for estimating weathering rates, their derivative nature reduces their value for assessing the mechanisms that control weathering rates, and they may not be suitable for use in predicting future weathering rates under conditions for which they

have not been calibrated or validated.

Taylor et al. (2011) build on a geochemical model developed by Banwart et al. (2009) which attempts to quantify the contribution of biologically derived protons and ligands to mineral weathering rates, and distinguishes between vegetation which forms arbuscular mycorrhizal associations and vegetation that forms ectomycorrhizal associations. Their model, based on the GEOCARBSULF model, assumes that AM fungi do not exude significant amounts of organic acids while

ectomycorrhizal fungi do, and models the activity of that exudation as that of oxalic acid. They also divide the soil volume into an area of immediate proximity to mycorrhizal hyphae, the mycorrhizosphere, and the bulk soil. When they applied their model over the last 200 Ma they observed that the draw-down of global atmospheric $CO_2$ levels over the last 120 Ma could largely be attributed to the emergence and diversification of angiosperms and the spread of ectomycorrhizal fungi. However, in addition to organic acid exudation, hyphal length density, which defined the volume of the mycorrhizosphere, was

parameterized to be 25-fold greater in ectomycorrhizal fungal dominated ecosystems than AM dominated systems, and modelled soil chemistry and the resulting terrestrial carbon sink, were highly sensitive to hyphal length density. Taylor et al. (2012) further developed this weathering model based on mycorrhizal association type and coupled it to a dynamic global vegetation model and validated it against a global dataset of watershed flux data. The resulting model, when applied over the last 200 Ma indicated that biological weathering was stronger in the distant past than today, and that vegetation and mycorrhizal

fungi have increased terrestrial weathering rates by a factor of 2. While their model performed reasonably well in the validation across a global series of catchment data, it did not support a distinct dichotomy in weathering behaviour between AM-dominated and EM-dominated ecosystems. Quirk et al. (2014) build on the model developments of Taylor et al. (2011, 2012) to illustrate the potential for a feedback between atmospheric $CO_2$ levels and biological weathering rates, such that, as $CO_2$ levels increase, global plant productivity and autotrophic soil inputs of protons and organic acids do as well, stimulating

biological weathering and serving as a negative feedback to increasing $CO_2$ levels, and as $CO_2$ levels decrease so does biological weathering. This sequence of models underlies important hypotheses concerning the role of land-plants in the geology of earth and the global biogeochemical carbon cycle, and incorporates the activity of weathering-promoting ligands and nutrient uptake, dynamic belowground allocation of carbon in response to plant nutrient stoichiometry (as a function of $[CO_2]$), discrete accounting for the fractional volume of soil intimately associated with roots and mycorrhizal hyphae, and a



potential framework to account for differential biological weathering activity of distinct vegetation types. While considerable evidence exists pointing to the potential for ectomycorrhizal fungi to be more potent weathering agents, than AM fungi, field studies comparing weathering rates in paired AM- and ectomycorrhiza- dominated forests have failed to find significant differences in mineral weathering rates (Koele et al., 2014; Remiszewski et al., 2016). Future applications utilising rhizosphere

or mycorrhizosphere vs bulk soil volumes should place more emphasis on the choice of hyphal length densities, and should likely use functions, as opposed to fixed parameters, that depend on plant type as well as plant productivity and nutrient status to describe fine root and mycorrhizal hyphal root lengths.

Roelandt et al. (2010) coupled a reactive transport model to the Lund-Potsdam-Jena global dynamic vegetation model, which they termed Biosphere-Weathering at the Catchment Scale (B-WITCH), and were able to model base cation efflux

accurately from the Orinoco watershed. They concluded that vegetation exerts a major role on mineral weathering rates, but that this role is primarily hydrological, via evapotranspiratory fluxes. However, while their model did feature organic ligand-promoted dissolution, the source of those ligands was decomposition only, they treat the entire rooting zone as a single interconnected solution, and while they do feature plant functional types, those functional types do not correspond to belowground physiology or mycorrhizal association. The B-WITCH model appears to reflect the most mechanistic approach

amongst global dynamic vegetation models to estimating mineral weathering rates, but additional processes may need to be implemented to capture the influence of biology on mineral weathering rates.

Maher et al. (2010) applied the reactive transport geochemical model CrunchFlow, which estimates weathering rates based on experimentally-derived dissolution equations for individual minerals, to examine the effect of fluid residence time, which in turn controls the transport of weathering products away from mineral surfaces, on mineral weathering rates. They observed

a strong inverse relationship between fluid residence time and weathering rates and interpreted this as clear evidence for transport control of weathering rates in natural ecosystems. Lawrence et al. (2014) coupled an organic acid module to the CrunchFlow model to examine the potential role of organic acids, modelled as oxalic acid, on mineral weathering rates, and observed that the primary effect of oxalic acid was to increase soluble Al but decrease free $Al^{3+}$ concentrations in solution; mineral weathering was enhanced near the zone of oxalic acid production (the topsoil) but decreased further down the profile

as rapid oxalate decomposition gave rise to elevated solution $Al^{3+}$ and increased secondary kaolinite precipitation, reducing overall weathering rates in the soil pedon. The authors acknowledge that more realistic, higher DOC values, may maintain Al in solution, eliminating this braking effect of oxalic acid, and that DOC character may not be accurately described by oxalic acid. Nevertheless the description of organic acid levels as the product of production and decomposition processes and the geochemical description of ligand-promoted chelation, dissolution, and transport may be useful process descriptions to model

the effects of biological exudates on mineral weathering rates and adaptable across a range of models. Winnick and Maher (2018) developed CrunchFlow to examine the dependence of mineral weathering rates on gaseous and dissolved $CO_2$ concentrations, and observed a very strong relationship between weathering rates and soil $CO_2$, and suggested that this may be an important mechanism by which soil respiration of vegetation (and mycorrhizal fungi) may stimulate mineral weathering.



Approaches to modelling mineral weathering by fungi have been discussed by Rosling et al. (2009). One major challenge is the choice of scale since fungal growth and function requires consideration of scales ranging from nano/micro-scale to kilometre scale and above where global biogeochemical phenomena are considered. In ectomycorrhizal systems the colonized root tips constitute the functional units where transfer of carbon to the fungus and nutrients to the plant occur. For the purpose

of modelling mycelial growth the root system is proposed as the model boundary from which newly colonized roots may emerge, but soil stratification and the distribution of different nutrients and different fungi between different soil horizons complicate the construction of models. Over 1000 species of ectomycorrhizal fungi occur in Swedish forests and we know that these species are not distributed evenly throughout the soil profile (Rosling et al., 2003; Sterkenburg et al., 2018) but we still know very little about the differences in physiology between individual species and how carbon allocation and nutrient uptake

are regulated.

Mobilization and eventual plant uptake of base cations may even be partially de-coupled from the concentrations measured in soil solution, complicating the use of these in models. More extreme extraction methods using high speed centrifugation may remove EPS from micropores and hyphal interfaces but the resulting bulk concentrations of weathering agents will not necessarily reflect those at active sites of weathering. Active uptake of weathering products by fungal hyphae, followed by

translocation towards the plant root, will prevent their accumulation at sites of weathering. Mineral elements mobilized by fungal hyphae may remain within the fungal mycelium for different lengths of time before becoming available for plant uptake and this may represent an important pool of base cations to be included in models, since ectomycorrhizal fungi cycle nutrients contained within both minerals and organic residues.

The stable isotope fractionation patterns of ectomycorrhizal fungi, shown by Fahad et al. (2016) involving discrimination

against heavier isotopes of Mg, provide a useful tool for use in future studies since they are possible to apply in field situations but further information about isotope fractionation patterns in organic and inorganic substrates is necessary since it is important to distinguish between the *de novo* supply of elements supplied via weathering of minerals and re-circulation of elements via decomposition of organic residues by both mycorrhizal and saprotrophic fungi.

## 8 Conclusions

In this paper we attempt to outline the consequences of interactions between minerals, microorganisms and plants at different spatial scales and to review the influence of biological processes on mineral weathering within an evolutionary context. The interaction of microorganisms with rocks and minerals must have been one of the earliest steps in the evolution of the different terrestrial ecosystems that we see today and there is documented evidence that early microorganisms had wide-ranging effects on both chemical and biological processes, including the accumulation of oxygen in the atmosphere, the evolution of over two

thirds of the minerals that exist today and the evolution of plastids through serial endosymbiosis. The subsequent evolution of higher plants made possible by efficient photosynthesis and successive increases in their size and ability to colonize and allocate photosynthetically-derived carbon to mineral (and organic) substrates, has enabled them to have increasing influence



as biogeochemical engineers (Fig. 11). Microbial symbionts have played an integral part in the evolution of plants and their ability to capture growth limiting nutrients such as N (Moreau et al., 2019). The influence of vegetation on mineral substrates is almost axiomatic but quantification of the contribution of plant-associated microorganisms to mineral weathering is problematical for two reasons. *Firstly* the ubiquitous distribution of microorganisms, the fact that plants devoid of microorganisms do not exist under natural conditions, means that plants need to be considered from a more holistic perspective, as holobionts, together with the many different microorganisms associated with them. *Secondly*, processes occurring at small spatial scales are difficult to quantify and upscale to the catchment, ecosystem or global scale. Although the combined effects of plants and their microbial symbionts have quantifiable effects on mineral dissolution and capture of nutrients, continued effort must be directed at elucidating the identity, distribution and functional characteristics of these many different microbial taxa. Additional information about their likely responses to different types of environmental stresses, including those induced by forest management practices, is an important research priority and some important knowledge gaps and questions are identified below. Weathering of minerals is important not just with respect to the sustainability of forestry. It is evident that the global weathering engine has had long-term effects on atmospheric $CO_2$ levels. Long term stabilization of C, derived from the atmosphere, in organic and mineral substrates, may take place through interactions involving glycoproteins, melanin, extracellular polymeric substances and formation of secondary minerals and mineraloids. Better understanding of these processes may facilitate improved forestry management practices that not only ensure sustainable production of biomass but can also be integrated into new CDR technologies.

### 9 Key questions and knowledge gaps

Better information on the identity of bacteria and fungi colonizing bedrock outcrops and other mineral substrates in forests

Forms of C sequestration in minerals in forests -how does weathering of silicates in forests eventually produce carbonates that are buried in the oceans?

Which fungal and bacterial taxa compete most successfully for plant derived C in the different mineral soil horizons

What are the effects of N deposition and fertilization (at different levels) on weathering and C sequestration?

### Author contributions

R.D. Finlay wrote most parts of the paper but with suggestions and inputs from all co-authors. S. Belyazid and N. Rosenstock, in particular, wrote most of the section on modelling and E. Bolou-Bi and H. Wallander provided substantial input to the section on stable isotopes. S. Mahmood provided substantial input with Figures 9 and 10. The work described in Figures 9 and 10 was carried out by S. Mahmood and Z. Fahad with advice from S. Köhler. E. Bolou-Bi carried out the stable isotope analyses in these experiments.



**Competing interests**

The authors declare that they have no conflict of interest.

**Acknowledgements**

This study was funded by the Swedish Research Council for Environment, Agricultural Sciences and Spatial Planning
FORMAS (grant no. 2011-1691) within the strong research environment "Quantifying weathering rates for sustainable
forestry (QWARTS)", with addition support to RDF from FORMAS (grant nos. 2014-01272 and 2017-00354).

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



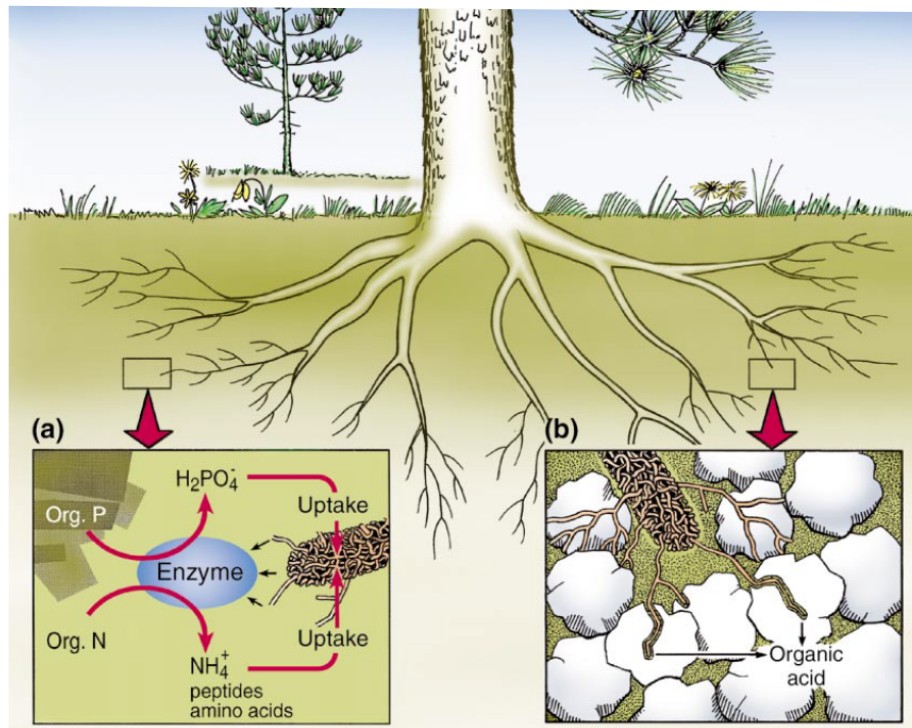

**Fig. 1.** Nutrient mobilization by mycorrhizal fungi, showing carbon allocation to-, and nutrent acquisition from-, organic and inorganic substrates. (Reproduced with permission; Landeweert et al.: Trends Ecol. Evol., 16, 248-254, 2002.)

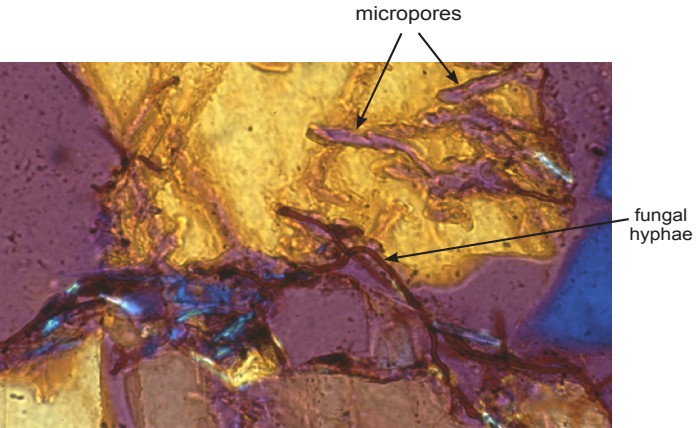

**Fig. 2.** Tubular pores in minerals, occupied by hyphae. Thin-section micrograph in cross-polarized visible light with 550-nm retardation (gypsum plate) showing an alkali feldspar from a podzol E soil horizon (yellow) with micropores (purple) and 5-mm thick fungal hyphae (brown). (Reproduced with permission; Jongmans et al.: Nature, 389, 682-683, 1997.)





**Fig. 3a**

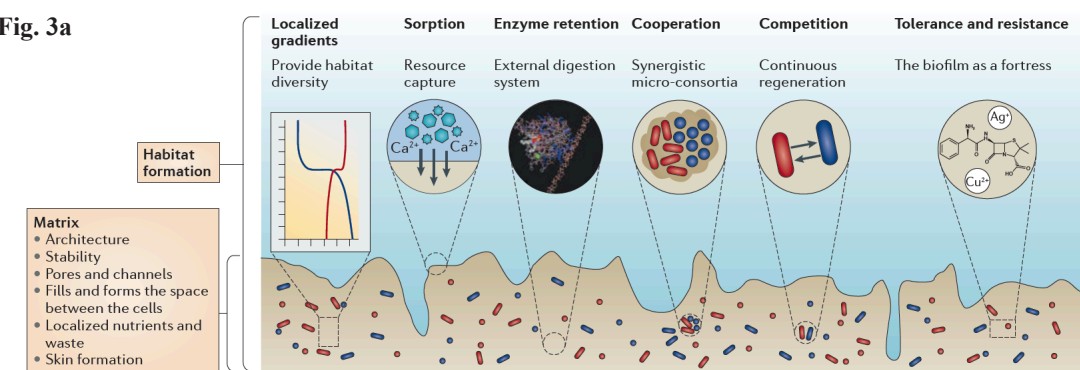

**Fig. 3b**

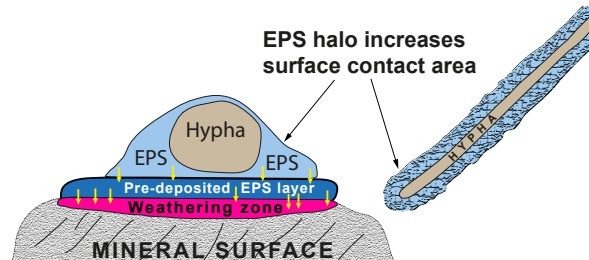

**Fig. 3.** (a) Schematic diagram showing biofilm structure and function, and the biological and chemical processes that biofilms influence. (b) Extracellular polymeric substance (EPS) halos and their possible influence on interactions between hypha and mineral surfaces (based on observations by Gazzè et al., 2013). (**Fig. 3a** reproduced with permission; Flemming et al.: Nat. Rev. Microbiol., 14, 563-575, 2016.)

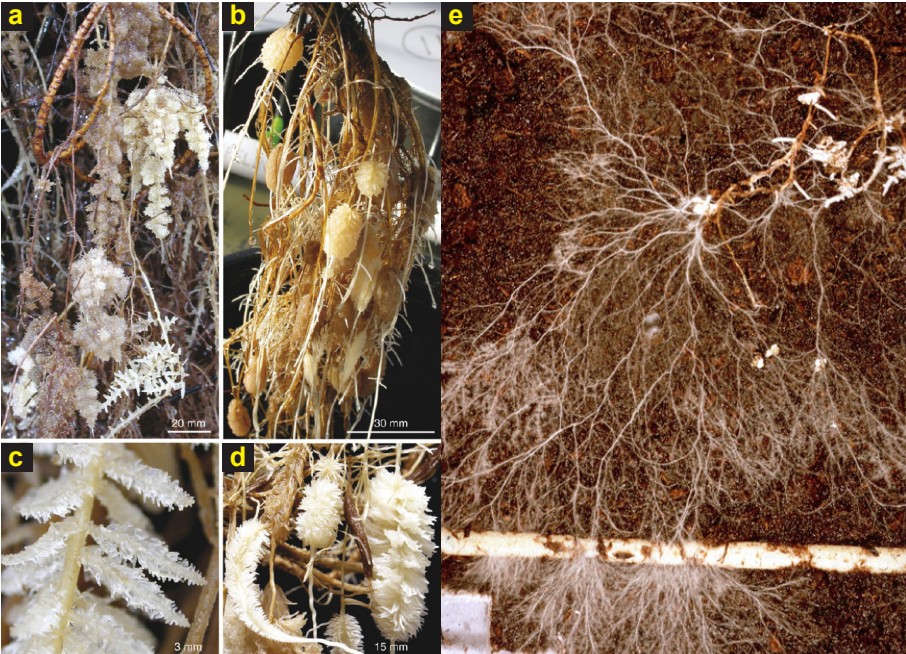

**Fig. 4.** Root morphology of Australian Proteaceae and South African Cyperaceae species grown hydroponically at extremely low P supply (≤ 1 μM). (**a**) *Dryandra sessilis* (Proteaceae) root system with 'compound' 'proteoid' root clusters. (**b**) *Hakea prostrata* (Proteaceae) root system with 'simple' 'proteoid' root clusters. (**c**) Compound proteoid root cluster of *Banksia grandis* (Proteaceae, Western Australia). (**d**) Simple proteoid root cluster of *Hakea sericea* (Proteaceae, eastern Australia). (**e**) shows ectomycorrhizal fungal mycelium of *Boletinus cavipes* extending from the roots of hybrid larch *Larix eurolepis*. (**a-d** Reproduced with permission; Lambers et al.: Trends Ecol. Evol., 23, 95-103, 2008.)
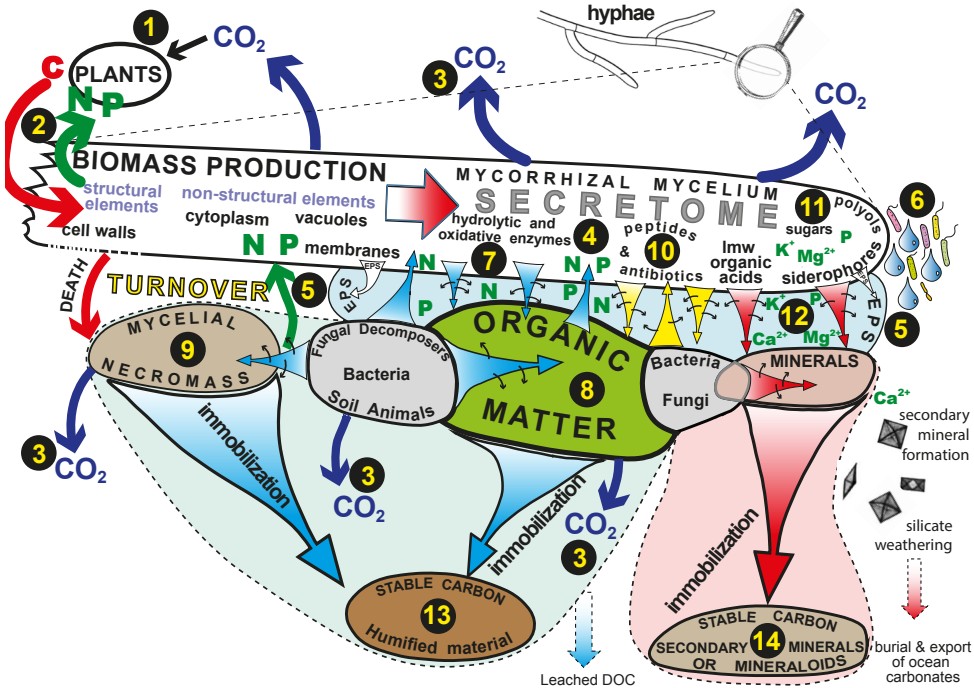

**Fig. 5.** The flow of plant-derived carbon through fungal hyphae to organic and inorganic substrates drives biogeochemical processes such as decomposition and weathering and influences patterns of C release and sequestration into stable organic and inorganic forms. Carbon is assimilated by plants (**1**) and transferred directly to symbiotic mycorrhizal hyphae that transfer nutrients mobilised by the hyphae back to their hosts (**2**). Products of mycelial respiration are released to the atmosphere (**3**). The fungal secretome (**4**) consists of different labile compounds that can be translocated to different organic or inorganic substrates. These compounds may be released into an extracellular polysaccharide matrix (**5**) or as droplets that condition the hyphosphere, facilitating interactions with bacteria (**6**). Hydrolytic and oxidative enzymes (**7**) mobilize N and P from plant-derived organic substrates (**8**) or microbial necromass (**9**). Peptides and antibiotics play important roles in signalling and influencing microbiome structure (**10**), sugars and polyols maintain osmotic gradients and hyphal turgor (**11**) and low molecular weight organic acids and siderophores influence the mobilization of P and base cations from minerals (**12**). Long term sequestration and stabilization of carbon can take place in recalcitrant organic substrates (**13**) and secondary minerals and mineraloids (**14**). (Reproduced with permission; Finlay, R. D. and Clemmensen, K.: Immobilization of carbon in mycorrhizal mycelial biomass and secretions, in: Mycorrhizal Mediation of Soil: Fertility, Structure and Carbon Storage, edited by: Johnson, N. C., Gehring, K., and Jansa, J.: Elsevier, Amsterdam. pp 413-440, 2017. ISBN:978-0-12-8043127)

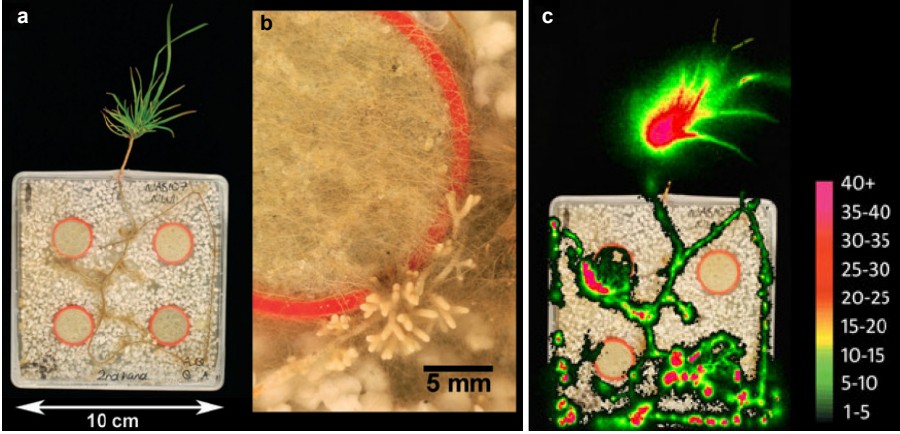

**Fig. 6.** (**a**) Sterile microcosm containing ectomycorrhizal *Pinus sylvestris* seedling colonized by *Paxillus involutus*, with plastic wells containing 10% (w/w) apatite in quartz sand (top left and bottom right) and two wells with quartz sand only (top right, bottom left). (**b**) Detail of an apatite containing well, colonized by *Paxillus involutus*. (**c**) digital autoradiograph showing $^{14}$C distribution 24 h after commencing labelling of shoots on a false colour scale of counts per 0.25 mm$^2$ pixel after imaging for 15 min. (Reproduced with permission; Smits et al.: Geobiology, 10, 445-456, 2012.)





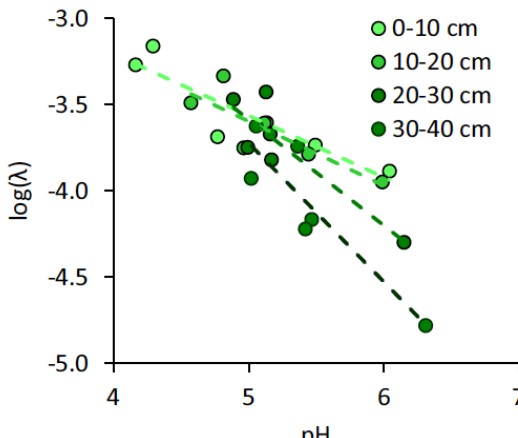

**Fig. 7.** Relationship between apatite loss due to weathering and pH, at different soil depths. (Reproduced with permission; Smits et al.: Plant Soil., 385, 217-228, 2014.)

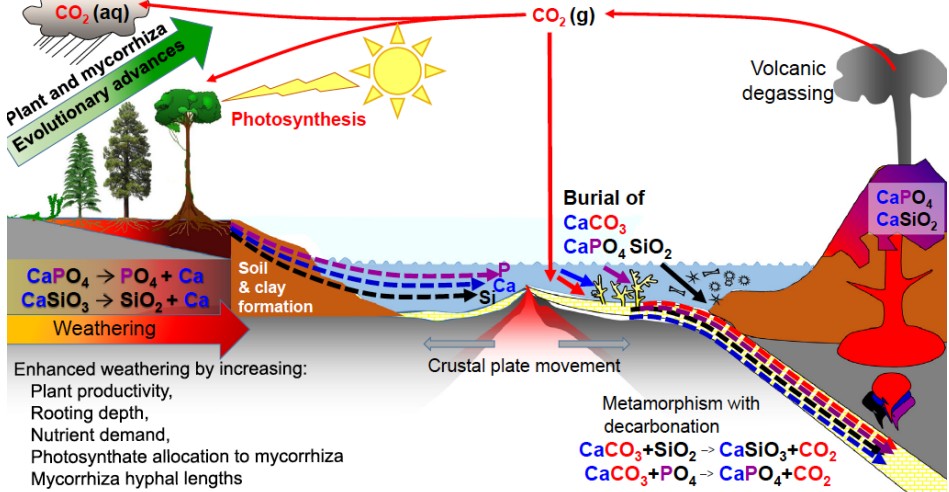

**Fig. 8.** The effects of evolutionary advancement in plants and mycorrhizas in the geochemical carbon cycle, increasing the weathering of calcium (Ca)-, phosphorus (P)-, and silicon (Si)-bearing minerals and generating clays. Plants and their mycorrhizal fungi have increased the rates of dissolution of continental silicates especially calcium silicate ($CaSiO_3$), and apatite (Ca phosphate-$CaPO_4$), but a portion of the Ca, P, and Si released from rocks is washed into the oceans where these elements increase productivity. Some of the Ca and P end up in limestone and chalk deposits produced by marine organisms such as corals and foraminifera, thereby sequestering carbon dioxide ($CO_2$) that was dissolved in the oceans into calcium carbonate ($CaCO_3$) rock for millions of years. Dissolved Si is used in sponges, radiolarians, and diatoms that can accumulate on the sea floor. The ocean sediments are recycled by subduction or uplift by tectonic forces, with volcanic degassing and eruptions of base-rich igneous rocks such as basalt returning Ca, P, Si, and other elements back to the continents, thereby reinvigorating ecosystems with new nutrient supplies through weathering. Note for simplicity that magnesium is not shown in the figure, but follows parallel pathways to Ca and is co-involved in sequestering $CO_2$ into dolomitic limestones. **aq**, liquid state; **g**, gaseous state. (Reproduced with permission: Leake, J. R., and Read, D. J.: Mycorrhizal Symbioses and Pedogenesis Throughout Earth's History. in: Mycorrhizal Mediation of Soil: Fertility, Structure and Carbon Storage. edited by: Johnson, N. C., Gehring, K., and Jansa, J.: Elsevier, Amsterdam. pp 9-33, 2017. ISBN:978-0-12-8043127)

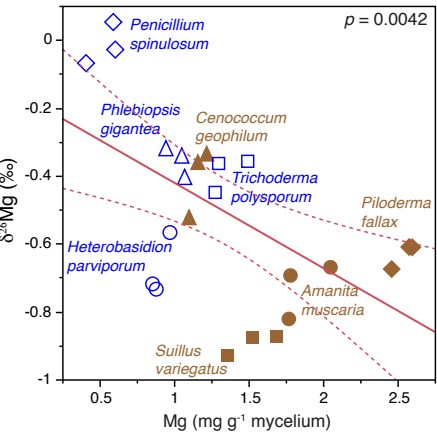

**Fig. 9.** Bivariate plots of $\delta^{26}Mg$ (‰) versus Mg concentration (mg g$^{-1}$) in mycelia of ectomycorrhizal and nonmycorrhizal fungi grown on mineral free Modified Melin-Norkrans (MMN) medium amended with granite particles. The fungi were grown on cellophane membranes covering the growth substrates in Petri dish microcosms. Open blue symbols represent nonmycorrhizal fungi and closed brown symbols represent ectomycorrhizal fungi. (Reproduced with permission; Fahad et al.: Env. Microbiol. Rep., 8, 956-965, 2016.)

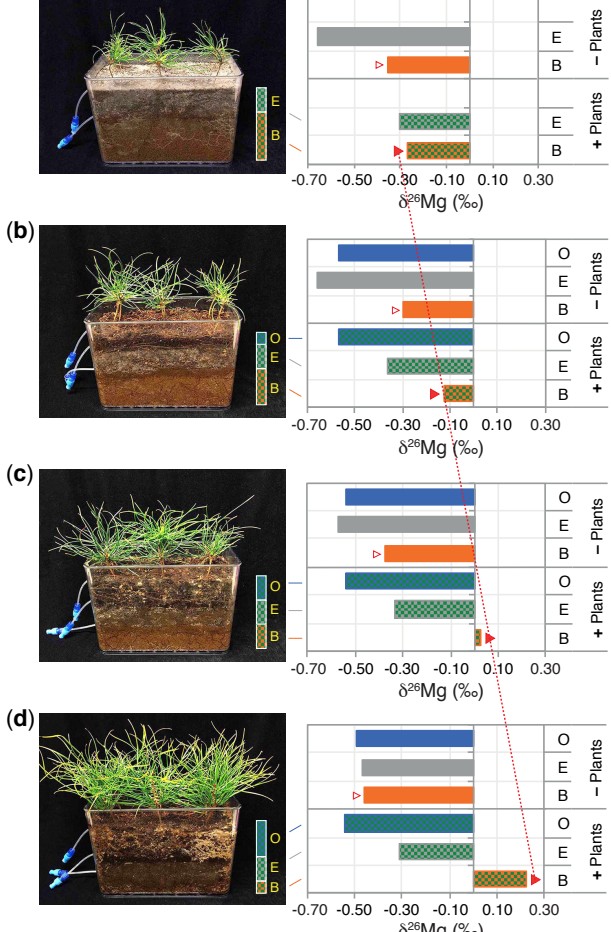

**Fig. 10.** Mesocosms containing reconstructed podzol soil profiles with different amounts of organic (O) horizon material to simulate different intensities of forest harvesting. (**a**) no O horizon, (**b**) 50% thickness O horizon, (**c**) Normal (100%) thickness O horizon, (**d**) 150% thickness O horizon. The histograms show levels of enrichment of $^{26}Mg$ in soil solution extracted from the O (organic), E (eluvial) and B (illuvial) horizons. The upper part of each diagram represents systems incubated without plants, the lower part of each diagram represents systems containing *Pinus sylvestris* seedlings (as illustrated). Note that the seedling growth is proportional to the amount of organic soil, from which ectomycorrhizal fungi mobilize N. The enrichment of $^{26}Mg$ in the soil solution becomes greater and greater with increasing plant growth (and therefore increasing Mg uptake) – but only in the B horizon, because there is discrimination against uptake of the heavy isotope. This suggests that the B horizon is the primary site of active mineral weathering and Mg uptake. Extensive colonization of roots, organic and mineral substrates by ectomycorrhizal mycelia is visible. Horizontal scale bar in (**a**) = 5 cm.





**Fig. 11.** Schematic diagram summarizing an evolutionary perspective of interactions involving mineral weathering and decomposition. Current rates of mineral weathering have been influenced by different 'events' and processes, including the effects of biological processes on mineral evolution (**1**), serial endosymbiosis (**2**), enabling the evolution of higher plants, mycorrhizal symbiosis (**3**) enabling increasing colonization of substrates by roots and mycorrhizal mycelium (**4**), leading to more efficient nutrient uptake and larger amounts of photosynthetic tissue (**5**). The evolution of ectomycorrhizal fungi (ECM) has enabled efficient extraction of N and P from recalcitrant organic material (**6**) powered by higher C allocation and better colonization of organic and mineral substrates (**6 & 7**).