# Peer review of "Biological weathering and its consequences at different spatial levels – from nanoscale to global scale"

_Biogeosciences, 2019_

## Referee Comment (RC1) · Thomas W. Kuyper (Referee) · 6 Mar 2019

Biological weathering and its consequences at different spatial levels – from  nanoscale to global scale

The ms is intended as a review of current progress made in understanding biological processes contributing to weathering. It is based on a very extensive reading in the literature (I counted 215 references) on a very wide range of topics, including chemistry, physics and biology, combining empirical and modelling approaches, based on a large range of experimental techniques. The authors also make a very laudable to attempt to scale up, both spatially (as indicated by the title) and temporally, when they link past and current weathering processes as an evolutionary and ecological force. The ms is also very well written. Despite all this initial praise, reading the ms did not fully satisfy me. In my view this is to a large extent due to the fact that it succeeds very well as a review, but succeeds to a lesser extent as a synthesis. Several empirical observations seem to contradict other observations, and one would like to read how much consensus has been reached on the biology of weathering. In that respect I found the final section (key questions and knowledge gaps) somewhat disappointingly short. Considering the lack of consensus on the importance or generality of several processes, a more cynical reader may easily be inclined to think that almost anything goes in biological weathering. In fact this kind of mild cynicism is almost encouraged by the authors: after having presented so many data the authors (p. 17, l. 33 – p. 18, l. 1; note that one of the authors of the paper referred to is also an author of this ms) state that "Smits & Wallander (2017) consider that there is *no clear evidence* [emphasis mine] that processes at the laboratory-scale play a significant role in soil-scale mineral dissolution rates". If so, what is the main message of this paper?

Let me try to back up my dissatisfaction with a couple of general observations. Before providing more detailed comments.

1.  The point of departure for the study is that weathering is the only or main supplier of base cations and phosphorus to compensate for losses through harvesting and leaching. However, on p. 17, l. 21 the reader is informed about atmospheric deposition (the only mention of this input source) where we are informed that a study found that atmospheric deposition was four times as important as weathering; and that the weathering flux was less than 0.3% of calcium uptake. This statement then raises questions about its quantitative importance over ecological time scales and evolutionary time scales, an issue treated very implicitly at best.

2.  Despite the generality of the title (biological weathering), the focus is almost exclusively on the role of mycorrhizal fungi plus associated mycorrhizosphere bacteria and the trees with which they associate (I like the focus on the plant as holobiont). Lichens, generally considering as major weathering agents in the first stages of primary succession, are mentioned only once (p. 12, l. 10-12). There the authors state that "the ubiquity and significance of lichens (..) as a model for understanding weathering (...) are well understood." However, the reader is not

informed about this understanding, nor is (s)he informed whether fungal weathering is similar or dissimilar from lichen weathering in any significant respect. There is also very limited attention for fungi other than mycorrhizal fungi, however from an evolutionary perspective this is a missed opportunity. A fungus often used in weathering studies is *Paxillus involutus*, a species derived from a clade of brown-rotting fungi characterized by oxalate production. It could be possible that the ability to produce and excrete oxalate in the environment evolved for different purposes and was even maintained in the ectomycorrhizal groups in this clade for different purposes.

3. There are many parts in the ms where the possible difference in weathering ability between arbuscular mycorrhizal fungi+plants and ectomycorrhizal fungi+plants are mentioned. Some of these are quite explicit in suggesting that the ectomycorrhizal symbiosis allows higher weathering rates than the arbuscular mycorrhizal symbiosis. However, we also learn that weathering evolved in the arbuscular mycorrhizal symbiosis (p. 11, l. 26) and that some studies did not find differences in weathering rates under ectomycorrhizal and arbuscular mycorrhizal vegetation (p. 21, l. 26-27). The reader of his paper will therefore remain in doubt what the current consensus view is (if there is consensus), what likely hypotheses exist to explain such different data and what kinds of research approaches exist to resolve that issue. (One option would be a common-garden experiment with sister clades of plants with the different mycorrhizal symbioses, in analogy of the approach by Koele et al. (New Phyt. 196: 845-852. 2012) when they tested for stoichiometric differences (leaf N:P ratio) between both guilds. I am sure there must be other ways to make progress as well.) Another group of mycorrhizal fungi + plants, which form the ericoid mycorrhizal symbiosis, is mentioned once (p. 7, l. 21) even though they have been suggested to be strong weathering agents as they can produce copious amounts of low-molecular-weight organic acids (Martino et al., Soil Biol. Biochem. 35: 133-141. 2003).

4. Addressing (and putatively answering) the question of the role of different mycorrhizal symbioses in weathering is, in my view, particularly relevant when it comes to understanding mechanisms. If weathering is driven by the production of LMWOA and siderophores, then it should be clear that the ectomycorrhizal symbiosis is much more important for weathering than the arbuscular mycorrhizal symbiosis (as AMF have not been reported to produce LMWOA, the AMF symbiosis has been reported to downregulate LMWOA production by plants (Ryan et al., Plant Cell Environ. 35: 2170-2180. 2012), and AMF do not produce siderophores as far as I know). If other mechanisms are more important (e.g., acidification driven by excess uptake of cations over anions and proton exudation to maintain charge balance; or dissolved $CO_2$ as a consequence of respiratory activity), the contribution by both guilds could be more important – with differences still related to the amount of extraradical hyphal biomass and / or respiratory activity.

5. The issue about the relative importance of weathering mechanisms has been debated since mycorrhizal researchers entered that field in the early 2000s. When

enthusiastic claims were made for a major role of mycorrhizal fungi (and I admit having been such an enthusiast as well), these ideas were criticised by Sverdrup, who essentially claimed that weathering was driven by CO2 flux and that the contribution by ectomycorrhizal fungi was around 2%. While his claim has been challenged (Van Schöll et al., Plant Soil 303: 35-47. 2008), I think this review would have been a good place to synthesise current understanding. Sverdrup (cited in the ms – pls note that the journal has Volume 23, Issue 4; not Volume 4) has maintained his suggestion about the major importance of respiration / $CO_2$ production, rather than the production of organic acids) as the driver for weathering, stating: "the growth of trees represents quantitatively largest single biological process that can affect weathering, followed closely be decomposition of organic matter." It is evident that the authors of this ms disagree with Sverdrup, however, without fully discussing this alternative view. I think this is a missed opportunity. The same applies to the origin of pores, with Sverdrup claiming that they are of abiotic origin (as cited in p. 3, l. 30). How would the authors of this ms evaluate our current knowledge and understanding? (Note that because of the extent of tunnelling the contribution to weathering might be limited, irrespective of the question on their origin.)

6. While I agree about the importance of upscaling, both spatially and temporally, I think that progress depends on the extent to which we can quantify rates. Unfortunately, the paper is quite frugal is giving numbers. This may give the impression that despite such many studies there has been little progress in quantifying processes. That conclusion seems also implied in p. 1, l. 31-32 ("opinion appears to be divided with respect to the quantitative significance [of interactions between microbes and minerals]"). If opinions are divided, please give equal hearing to arguments from both sides. But if a clearer picture has emerged in the view of the authors, please provide more quantitative detail. In order to have feedback mechanisms to work over both ecological and evolutionary times, we need such data.

7. The authors refer (p. 2, l. 28) to twelve testable hypotheses on the geobiology of weathering. If would help the reader to list those (rather than to invite them to look up the paper themselves) and to indicate to what extent their review helps addressing these hypotheses. For instance hypothesis 8 (elevated $CO_2$ will enhance weathering) seems to assume that weathering fluxes and its ultimate consequences of drawdown of $CO_2$ occur at very different time scales, which could put constraints on feedback mechanisms postulated in the ms. With respect to hypothesis 2, the importance of stoichiometry, I think that the studies done of mycorrhizal weathering provide much of the needed data. In none of their hypotheses they draw attention to different kinds of mycorrhizal symbiosis (but this could be a refinement of hypothesis 1), although it may not be coincidental that their figure 1 depicts an ectomycorrhizal conifer...

Some of these comments will make the manuscript longer, so I think it may help if I indicate cases were shortening of the ms is possible. I noted several digressions (also in the Abstract) that result in a less focused paper. Examples are: reference to acquisition of N and P by mechanisms other than weathering (p. 7, l. 18-34), hydraulic lift / redistribution (p. 9, l. 6-8), oxygenic and anoxygenic photosynthesis (p. 11, 14-24), autotrophic and heterotrophic respiration in forests (p. 13, l. 30 – p. 14, l. 6; unless the authors think that Sverdrup is, essentially, right...), differential carbon storage in ectomycorrhizal and arbuscular mycorrhizal forests (p. 14, l. 4-28; note that of the two biomes where both guilds occur larger C storage per unit N was shown for the temperate biome, not for the (sub-)tropical biome – so we should not take Averill's claim too seriously), nitrogen in the rhizosphere (p. 24, l. 1-2).

Page-by-page comments

p. 3, l. 9    Is the Finlay & Clemmensen paper on biogenic weathering? The title of the paper would suggest otherwise

p. 5, l. 14    Please provide a reference for the suggestion about the importance of horizontal gene transfer in such microbial consortia in EPS.

p. 7, l. 1    Here I disagree. In such habitats, in case of a low pH, plants with cluster roots (or proteoid roots; I think they are the same) or dauciform roots produce carboxylates that *desorb* phosphorus from mineral surfaces. But desorption is not weathering, dissolution of minerals. Weathering would happen in the case of high-pH with calcium phosphates; in low-pH soils P is far too scarce to form substantial amounts of Fe- and Al-phosphates that are weathered.

p. 9, l. 14    In the light of current criticisms of humic and fulvic acids as large molecules (Lehmann & Kleber claim these to be aggregates of essentially small molecules) this statement may need reconsideration in terms of underlying mechanisms.

p. 15, l. 21    When introducing the Blum et al. hypothesis, they should also refer to contradictory data by Dijkstra and Smits (now only referred to on p. 17, l. 18-22; however I interpret that paper as showing that Blum et al.'s conclusion is grossly overstated – but I would love to see the opinion of the authors of this ms).

p. 17, l. 19    Please provide a reference to that further study.

p. 19, l. 17    Note that exudation of carboxylates / organic anions can also have a major function in the desorption of iron-oxide bound soil organic matter and the subsequent acquisition of carbon (Keiluweit et al., Nature Clim. Change 5: 588-595. 2015) and nitrogen (Jilling et al., Biogeochemistry 139: 103-122. 2018) (and possibly phosphorus, as both inorganic and organic P are sorbed on such surfaces).

Thomas W. Kuyper

---

## Referee Comment (RC2) · Anonymous Referee #2 · 26 Mar 2019

In General:

The authors of the paper propose a review of current (last 10 years) of advancement made in the understanding of biological weathering, specifically focusing on the boreal forest, in response to an interdisciplinary project called "Quantifying weathering rates for sustainable forestry." This topic fits well the scope of BG and it is needed to help us move forward in this area of research.

The manuscript cites more than 200 references that span a wide range of topics from physical, chemical and biological approaches, and scales from nano-scale empirical studies to global scale modeling, and emphasizing an evolutionary viewpoint on biolog-

ical weathering. However, I was disappointed seeing that about 1/3rd of the references are prior to 2009 and have been widely cited and reviewed in the past, thus these do not give "anything new" especially in some sections of the manuscript (see details below) and it also contradicts with the authors aim of summarizing the last 10 years of advancement.

The manuscript is a well-written review/summary of more than the last 10 years of biological weathering research with a heavy emphasis on mycorrhizae mediated weathering (which is the ecosystem in the boreal forest). However, there is no synthesis of the reviewed literature, there is no agreement stated on what is the current understanding, or state of this biological weathering in the boreal forest, and how it applies to sustainable forestry or simply how to move forward. The manuscript is a review, but it lacks a synthesis.

Regardless of great writing, it was not an easy read, because I could not find/follow the purpose of this manuscript, it presents a lot of data on both side of the arguments that contradict each other, which is fine, however, there are no directions, there is a lot of rambling on without focus – what is the underlying message? What do the authors want to achieve with this review? Key questions and knowledge gaps section is underdeveloped and it seems like it was an afterthought and stuck to the end.

In addition, I think that the title is misleading, as the review is really about weathering in the boreal forest. Most cited work was done by researchers related to the boreal or other forests (field), in the laboratory using mostly conifers and mycorrhizal fungi, and there are couple of "side topics" that seems to be out of place in this bigger scheme (for example, the hydraulic lift study for drought-prone ecosystems).

Some specifics:

Abstract and 1. Introduction – no specific comments.

2. Microscale/nanoscale observations of physical alteration of minerals: This section is

heavily based on older findings and mention some new studies, but it is unclear what advances were made in the last 10 years – new techniques? New understanding of processes? Or just supporting previous findings? Or all above? It needs a refocus, and it can be shortened by about half and still convey the same message.

3. Biofilms and small scale microbial interactions with consequences at higher spatial scale: how are these differently categorized than the next section, which is about microbial and plant secretions? EPS, biofilm, oxalic acids etc. are secretions, are not? What are the consequences at higher spatial scale? Do we know? Or is it a challenge to scale things up? Again, what is the new advancement in the last 10 years? The section needs some clarifications and/or refocus.

4. Microbial and plant secretions – evidence from microcosms and mesocosms: long section – rambling on without focus, lots of info and data about various roles, functions, and processes of mycorrhizal fungi, but no other components of the ecosystem, and the hydraulic lift section seems irrelevant in the boreal forest. Bringing in drought may be something we want to think about as climate shifts, but it most likely causing larger problems in drought-prone parts of the world.

5. Systemic consequences of microorganism-mineral interactions in an ecological and evolutionary context: this is really important and interesting, however, it is too long, have some repetition – I am not sure why the 5.1. section is separated (elevated) from the rest of 5. – Weathering, nutrient acquisition, carbon allocation, and sequestration are the key elements of the evolutionary viewpoint – perhaps, this section could be rearranged and shortened to synthesize our current understanding of the evolution of plants and associated fungi in the context of carbon and nutrient cycling. Bob Berner did the pioneering work in this field with his carbon models, but it got a lot of attention in the last 10 years, so a focused synthesis would help us to identify future directions.

6. Methods using stable isotopes: The section is interesting, provide laboratory evidence of the usefulness of these techniques in addition to field studies, however, the

last paragraph states that the "there is no clear evidence that processes observed at the laboratory-scale play a significant role in "soil-scale" mineral dissolution rates." This indicates that laboratory studies are useless, why do we bother then? Is there anything we learned from the laboratory studies? Also, the last paragraph is a repetition of statements on page 12 lines 13-15.

7. Modelling of weathering in forest soils: this whole section is unfocused. It starts with the PROFILEand ForSAFE models, then it talks about information needs and possible improvements (in 7.1.) and then it returns to talk about a bunch of other models in too much detail without getting to a point. This section should synthesize what are the main outcomes of the different modeling approaches (probably in half of the length), and identify what is missing (information) and how to tackle the shortcomings.

8. Conclusions: I was expecting to find the key questions, knowledge gaps and future directions (or call for specific areas of research) in this section.

Figures: Not all necessary – Figure 1, 2, 4, 6, 7 do not add new information to the summary (synthesis) or not necessary to understand the text. Figure 9 and 10 are a good representation of specific examples for laboratory approaches. Figure 3, 5, 8, and 11 are great illustrations of processes and their interactions from small to large scales.

---

## Referee Comment (RC3) · Anonymous Referee #3 · 29 Apr 2019

General comments:

Finley and coauthors provide here an interesting and timely review on biological weathering across scales. It is well written and meets current questions and gaps of knowledge in this field. The general organization of the manuscript might on the other hand be significantly improved. I do not doubt however that some restructuring will enable this discussion paper to reach a wide audience and the large impact it deserves.

I would first like to acknowledge the fact that covering such a wide topic is chellenging, and I would like to congratulate the authors for their effort to try to bring together various aspects of the study of biological weathering in one single review paper. In that

respect, I found the general organization according to spatial scales very attractive in the first place. The resulting sections, however, lack of focus, while the last sections do not seem to follow this original plan (e.g. section 6 on insights from stable isotope methods). As a result, the reader might get easily lost or distracted by some of the digressions.

I think that the richness of ideas and concepts gathered here is a real originality of this review, but the author may want to be careful that the reader keeps track of the point that they are trying to make in a given paragraph. Section 5, which gathers a main section introducing concepts as diverse as "mineral evolution", the geological carbon cycle or plants as holobionts and another subsection on carbon allocation and sequestration including carbon cycle and geoengineering concepts is for instance a little hard to digest.

To improve this point, I could first suggest gathering the different processes and links existing between them in a dedicated introductory section to make sure all readers are on the same page before tackling more detailed aspects of each scale. For instance, the relevance of allusions to long-term sequestration of carbon (e.g. lines 7-9 p. 8 and lines 24-26 p. 6) for the general topic of the paper might be unclear to some readers until they reach section 5.1. Another example is the geological cycle of carbon, the presentation of which is scattered across section 5 and somewhat redundant (e.g. p. 11 and 14). An introductory section could also enable to present the order of magnitude of the different processes and elemental fluxes to be considered here (e.g. typical elemental flux derived from primary mineral weathering vs. typical plant uptake and potential export related to forestry practices vs. typical atmospheric input for a given type of system) which is something missing here. Second, I would recommend organizing sections into subsections to keep the reader oriented. I would also avoid sections including a sort of single small subsection, e.g. 5->5.1->6 or 7->7.1->8.

Another general point is that I find that the manuscript is lacking a few but quite important references. I try to provide a couple of them in the specific comments section

below, which I hope the authors will find helpful. Aside from those points, I am enthusiastic about this interesting manuscript and I would recommend its publication provided that a couple of modifications and restructuring are done.

Specific comments:

-Section 2: Alt and Mata (2000), Benzerara et al. (2007), Furnes et al. (2001) and Torsvik et al. (1998) are additional references on the biotic origin of tubular structures that the authors might find useful to include. l.12 p.4: the effect of turgor pressure on biomineral weathering is also discussed by Li et al. (2016)

-Section 3: Maybe the first paragraph might be strengthened by adding a couple of references when presenting common biofilm features to guide the reader, especially if some studies are relating these biofilm properties (e.g. retention of water) to mineral weathering (e.g. fluid-mineral contact time). In the second paragraph, Barker et al. (1998) is probably another classical reference on biofilms and microenvironments that might be added. In the last paragraph dealing with the interplay between bacteria and mineral weathering should be strengthened in my opinion. Some recent references including Mitchell et al. (2013), Montross et al. (2013), Wild et al. (2018) and Wild et al. (2019) are missing here and should be included at this point I think. l.19 p.6: "Burial" is referred to as "incubation" in Uroz et al. (2012). I would recommend sticking to this latter term. l.23 p.6: I am not completely sure of the relevance of the position of the last sentence (l. 23-26). I would move it upward or delete it.

-Section 4: l.29 p.6: the statements of the production of acidifying substances (H+, organic acids) and ligands that complex with metals in the minerals may need to be supported by quotations. l.30 p.6:: "that retard weathering rates" reduce or decrease weathering rates would be more accurate l.7 p.7: "uptake of positively charged nutrients such as NH4+ and K+, result in exudation of protons" may benefit from the support of a quotation.

-Section 5: This section is a little bit dense, I would suggest dividing it into subsections.

[Figure]

-Section 6: This section is thematic, not intrinsically associated to a given scale. Also, I am questioning the scientific relevance of specifically distinguishing studies from the QWARTS project from other studies.

-Section 7: Direct in situ measurements using gravimetric approaches by Augusto et al. (2000) or Turpault et al. (2009) or interferometry methods by Wild et al. (2019) are not reported by Akselsson et al. (2019) but might be worth mentioning since they directly meet some of the challenges implicitly pointed out in this manuscript regarding the validation of weathering models and the transposition/upscaling of laboratory mesocosms to field systems. In the second paragraph, I find the description of the influence of the different processes on the dissolution rate a little bit unclear, and I feel that the clarity of this section might be improved. Otherwise, readers who are not familiar with that type of models will be easily lost. I would suggest reorganizing this section and starting by presenting the different parameters controlling the dissolution rate (temperature, pH, chemical affinity, ...) and then, in a second step, describing the influence of plant metabolism on these factors and thus on the dissolution rate. I would also strongly recommend using an equation (e.g. developed from equation 3 in Erlandsson et al. (2016), equation 3 in Godderis et al. (2006) or equation 1 in Palandri and Kharaka (2004)) to visually support this discussion. I would also avoid mentioning the concepts of "weathering brakes" or "transition state theory" if they are not explained. This might be more confusing than useful for readers, depending on their background.

References:

Alt, J.C., Mata, P., 2000. On the role of microbes in the alteration of submarine basaltic glass: a TEM study. Earth and Planetary Science Letters 181, 301-313.

Augusto, L., Turpault, M.P., Ranger, J., 2000. Impact of forest tree species on feldspar weathering rates. Geoderma 96, 215-237.

Barker, W.W., Welch, S.A., Chu, S., Banfield, J.F., 1998. Experimental observations of the effects of bacteria on aluminosilicate weathering. Am. Miner. 83, 1551-1563.

Benzerara, K., Menguy, N., Banerjee, N.R., Tyliszczak, T., Brown, G.E., Guyot, F., 2007. Alteration of submarine basaltic glass from the Ontong Java Plateau: A STXM and TEM study. Earth and Planetary Science Letters 260, 187-200.

Erlandsson, M., Oelkers, E.H., Bishop, K., Sverdrup, H., Belyazid, S., Ledesma, J.L.J., Köhler, S.J., 2016. Spatial and temporal variations of base cation release from chemical weathering on a hillslope scale. Chem. Geol. 441, 1-13.

Furnes, H., Muehlenbachs, K., Torsvik, T., Thorseth, I.H., Tumyr, O., 2001. Microbial fractionation of carbon isotopes in altered basaltic glass from the Atlantic Ocean, Lau Basin and Costa Rica Rift. Chem. Geol. 173, 313-330.

Godderis, Y., Francois, L.M., Probst, A., Schott, J., Moncoulon, D., Labat, D., Viville, D., 2006. Modelling weathering processes at the catchment scale: The WITCH numerical model. Geochim. Cosmochim. Acta 70, 1128-1147.

Li, Z.B., Liu, L.W., Chen, J., Teng, H.H., 2016. Cellular dissolution at hypha- and spore-mineral interfaces revealing unrecognized mechanisms and scales of fungal weathering. Geology 44, 319-322.

Mitchell, A.C., Lafreniere, M.J., Skidmore, M.L., Boyd, E.S., 2013. Influence of bedrock mineral composition on microbial diversity in a subglacial environment. Geology 41, 855-858.

Montross, S.N., Skidmore, M., Tranter, M., Kivimaki, A.L., Parkes, R.J., 2013. A microbial driver of chemical weathering in glaciated systems. Geology 41, 215-218.

Palandri, J.L., Kharaka, Y.K., 2004. A compilation of rate parameters of water-mineral interaction kinetics for application to geochemical modeling, in: Survey, U.S.G. (Ed.), U.S. Geological Survey, Open File Report. U.S. Geological Survey, Open File Report, p. 70.

Torsvik, T., Furnes, H., Muehlenbachs, K., Thorseth, I.H., Tumyr, O., 1998. Evidence for microbial activity at the glass-alteration interface in oceanic basalts. Earth and

Planetary Science Letters 162, 165-176.

Turpault, M.-P., Nys, C., Calvaruso, C., 2009. Rhizosphere impact on the dissolution of test minerals in a forest ecosystem. Geoderma 153, 147-154.

Uroz, S., Turpault, M.P., Delaruelle, C., Mareschal, L., Pierrat, J.C., Frey-Klett, P., 2012. Minerals affect the specific diversity of forest soil bacterial communities. Geomicrobiology Journal 29, 88-98.

Wild, B., Daval, D., Beaulieu, E., Pierret, M.-C., Viville, D., Imfeld, G., 2019. In-situ dissolution rates of silicate minerals and associated bacterial communities in the critical zone (Strengbach catchment, France). Geochim. Cosmochim. Acta 249, 95-120.

Wild, B., Imfeld, G., Guyot, F., Daval, D., 2018. Early stages of bacterial community adaptation to silicate aging. Geology 46, 555-558.

---

## Author Comment (AC1) · 20 May 2019

**Reply to Reviewer 1**

**Biological weathering and its consequences at different spatial levels – from nanoscale to global scale**

*Reviewer 1 makes a number of helpful comments that will no doubt greatly improve our manuscript if we are permitted to submit a revised version.*

The ms is intended as a review of current progress made in understanding biological processes contributing to weathering. It is based on a very extensive reading in the literature (I counted 215 references) on a very wide range of topics, including chemistry, physics and biology, combining empirical and modelling approaches, based on a large range of experimental techniques. The authors also make a very laudable to attempt to scale up, both spatially (as indicated by the title) and temporally, when they link past and current weathering processes as an evolutionary and ecological force. The ms is also very well written.

*We are pleased the reviewer considers the text well written.*

Despite all this initial praise, reading the ms did not fully satisfy me. In my view this is to a large extent due to the fact that it succeeds very well as a review, but succeeds to a lesser extent as a synthesis. Several empirical observations seem to contradict other observations, and one would like to read how much consensus has been reached on the biology of weathering. In that respect I found the final section (key questions and knowledge gaps) somewhat disappointingly short.

*We agree fully that the final section is too short and intend to expand it. (see below).*

Considering the lack of consensus on the importance or generality of several processes, a more cynical reader may easily be inclined to think that almost anything goes in biological weathering. In fact this kind of mild cynicism is almost encouraged by the authors: after having presented so many data the authors (p. 17, l. 33 – p. 18, l. 1; note that one of the authors of the paper referred to is also an author of this ms) state that "Smits & Wallander (2017) consider that there is *no clear evidence* [emphasis mine] that processes at the laboratory-scale play a significant role in soil-scale mineral dissolution rates". If so, what is the main message of this paper?

*We do not agree that the lack of consensus reflects "cynicism". **Indeed, one of the aims of this article was to examine differences in the degree of consensus about research conducted at different temporal and spatial scales.** (Admittedly this is not very well explained at present, but one of the take-home messages was meant to be that there is greater consensus on the large-scale, systemic effects of microorganisms on weathering than on the overall significance of micro-/nano-scale observations). Science proceeds by first identifying conflicting opinions and then, often at a later stage, resolving conflicts by collecting additional data – often in newly designed experiments or by using new techniques. It would be surprising if the **eleven** authors of this article had exactly the same opinion about every single process discussed, but we do agree that we should have worked much harder to try to resolve conflicting results to provide the "**synthesis**" the reviewer wants. At any one point in time it is often not possible to resolve conflicts of information completely but, in that case we agree that it is important to identify new approaches or key questions to ask in new experiments and to provide clear guidance about the approaches required. We will re-write the text to improve this aspect and expand the final section – as suggested above.*

Let me try to back up my dissatisfaction with a couple of general observations. Before providing more detailed comments.

1. The point of departure for the study is that weathering is the only or main supplier of base cations and phosphorus to compensate for losses through harvesting and leaching. However, on p. 17, l. 21 the reader is informed about atmospheric deposition (the only mention of this input source) where we are informed that a study found that atmospheric deposition was four times as important as weathering; and that the weathering flux was less than 0.3% of calcium uptake. This statement then raises questions about its quantitative importance over ecological time scales and evolutionary time scales, an issue treated very implicitly at best.

*We agree that alternative sources of different nutrients and base cations should be discussed and that there may be significant input under some circumstances from atmospheric deposition – examples of significant P input from atmospheric deposition to coastal Fynbos systems (eg. Brown et al. 1984) and the Florida everglades (Redfield, 2002) have been shown. These possible alternatives are now included in our discussion. However, this does not call into question the validity of all weathering studies and the clear stable isotope results from our own mesocosm experiment suggest that in boreal forest soils mobilization of Mg is probably not primarily from litter re-cycling in surface soil as shown by Dijkstra and Smits (2002) for Ca.*

2. Despite the generality of the title (biological weathering), the focus is almost exclusively on the role of mycorrhizal fungi plus associated mycorrhizosphere bacteria and the trees with which they associate (I like the focus on the plant as holobiont). Lichens, generally considering as major weathering agents in the first stages of primary succession, are mentioned only once (p. 12, l. 10-12). There the authors state that "the ubiquity and significance of lichens (..) as a model for understanding weathering (...) are well understood." However, the reader is not informed about this understanding, nor is (s)he informed whether fungal weathering is similar or dissimilar from lichen weathering in any significant respect. There is also very limited attention for fungi other than mycorrhizal fungi, however from an evolutionary perspective this is a missed opportunity. A fungus often used in weathering studies is *Paxillus involutus*, a species derived from a clade of brown- rotting fungi characterized by oxalate production. It could be possible that the ability to produce and excrete oxalate in the environment evolved for different purposes and was even maintained in the ectomycorrhizal groups in this clade for different purposes.

*Apart from ectomycorrhizal fungi in forests we do also mention **1.** proteoid roots of in highly weathered soils, **2.** calcicole plants in calcareous soils, **3.** non-mycorrhizal fungi such as different Aspergillus species and **4.** different bacterial species. However, we agree that a slightly better description of the potential role of bacteria, lichens and non-mycorrhizal fungi as weathering agents should be included and will include more information in the revised version of the manuscript. This will include evolutionary aspects (discussed by Fahad et al., 2016) and also the desired information about Paxillus involutus – recently discussed by Nicholás et al 2019. (ISME J. 13: 977-988).*

3. There are many parts in the ms where the possible difference in weathering ability between arbuscular mycorrhizal fungi+plants and ectomycorrhizal fungi+plants are mentioned. Some of these are quite explicit in suggesting

that the ectomycorrhizal symbiosis allows higher weathering rates than the arbuscular mycorrhizal symbiosis. However, we also learn that weathering evolved in the arbuscular mycorrhizal symbiosis (p. 11, l. 26) and that some studies did not find differences in weathering rates under ectomycorrhizal and arbuscular mycorrhizal vegetation (p. 21, l. 26-27). The reader of his paper will therefore remain in doubt what the current consensus view is (if there is consensus), what likely hypotheses exist to explain such different data and what kinds of research approaches exist to resolve that issue. (One option would be a common-garden experiment with sister clades of plants with the different mycorrhizal symbioses, in analogy of the approach by Koele et al. (New Phyt. 196: 845-852. 2012) when they tested for stoichiometric differences (leaf N:P ratio) between both guilds. I am sure there must be other ways to make progress as well.) Another group of mycorrhizal fungi + plants, which form the ericoid mycorrhizal symbiosis, is mentioned once (p. 7, l. 21) even though they have been suggested to be strong weathering agents as they can produce copious amounts of low-molecular-weight organic acids (Martino et al., Soil Biol. Biochem. 35: 133-141. 2003).

*We agree that some of these ideas are not currently included and will mention them in the revised manuscript. One of the arguments about possible differences between arbuscular mycorrhizal and ectomycorrhizal fungi concerns evolutionary differences of the C-fixation properties of their plant hosts (work of KJ Field and D Cameron) and these ideas will be discussed in the revised manuscript.*

4. Addressing (and putatively answering) the question of the role of different mycorrhizal symbioses in weathering is, in my view, particularly relevant when it comes to understanding mechanisms. If weathering is driven by the production of LMWOA and siderophores, then it should be clear that the ectomycorrhizal symbiosis is much more important for weathering than the arbuscular mycorrhizal symbiosis (as AMF have not been reported to produce LMWOA, the AMF symbiosis has been reported to downregulate LMWOA production by plants (Ryan et al., Plant Cell Environ. 35: 2170-2180. 2012), and AMF do not produce siderophores as far as I know). If other mechanisms are more important (e.g., acidification driven by excess uptake of cations over anions and proton exudation to maintain charge balance; or dissolved $CO_2$ as a consequence of respiratory activity), the contribution by both guilds could be more important – with differences still related to the amount of extraradical hyphal biomass and / or respiratory activity.

*This is an interesting subject area and there are probably questions that cannot be fully resolved with currently available information but we will add some comments. Generalisations should always be made with care but one aspect that could be relevant is differences in decomposition rates. Since ectomycorrhizal (and ericoid) fungi typically dominate systems characterized by recalcitrant organic substrates and slow decomposition rates, whereas AM fungi dominate systems with higher decomposition rates in which the input from turnover of nutrients from organic residues may be higher. Possible future approaches include common-garden experiments, as mentioned above, as well as the use of mutant plants with altered regulation of proton-pumping and some of these ideas will now mentioned in the revised manuscript.*

5. The issue about the relative importance of weathering mechanisms has been debated since mycorrhizal researchers entered that field in the early 2000s.

When enthusiastic claims were made for a major role of mycorrhizal fungi (and I admit having been such an enthusiast as well), these ideas were criticised by Sverdrup, who essentially claimed that weathering was driven by $CO_2$ flux and that the contribution by ectomycorrhizal fungi was around 2%. While his claim has been challenged (Van Schöll et al., Plant Soil 303: 35-47. 2008), I think this review would have been a good place to synthesise current understanding. Sverdrup (cited in the ms – pls note that the journal has Volume 23, Issue 4; not Volume 4) has maintained his suggestion about the major importance of respiration / $CO_2$ production, rather than the production of organic acids) as the driver for weathering, stating: "the growth of trees represents quantitatively largest single biological process that can affect weathering, followed closely be decomposition of organic matter." It is evident that the authors of this ms disagree with Sverdrup, however, without fully discussing this alternative view. I think this is a missed opportunity. The same applies to the origin of pores, with Sverdrup claiming that they are of abiotic origin (as cited in p. 3, l. 30). How would the authors of this ms evaluate our current knowledge and understanding? (Note that because of the extent of tunnelling the contribution to weathering might be limited, irrespective of the question on their origin.)

*We have discussed the fact that tunnelling may be both biotic and abiotic and that there are ways of distinguishing the two types of tunnels. ALL tunnelling is not abiotic. However we also discuss the important studies of Smits et al. (2005) that showed that tunnelling is not quantitatively significant as the sole indicator of weathering. We now mention the review by Van Schöll et al. (2008), as well as many other more recent studies. In fragmented mineral substrates weathering of surfaces may take place without formation of tunnels. We do fully understand that the growth of trees affects weathering of minerals and have explained that removal of weathering products from these sites (by hyphae) is an important process.*

6. While I agree about the importance of upscaling, both spatially and temporally, I think that progress depends on the extent to which we can quantify rates. Unfortunately, the paper is quite frugal is giving numbers. This may give the impression that despite such many studies there has been little progress in quantifying processes. That conclusion seems also implied in p. 1, l. 31-32 ("opinion appears to be divided with respect to the quantitative significance [of interactions between microbes and minerals]"). If opinions are divided, please give equal hearing to arguments from both sides. But if a clearer picture has emerged in the view of the authors, please provide more quantitative detail. In order to have feedback mechanisms to work over both ecological and evolutionary times, we need such data.

*We agree that more quantitative estimates (rates) are necessary and will attempt to cite more quantitative estimates in the revised manuscript..*

7. The authors refer (p. 2, l. 28) to twelve testable hypotheses on the geobiology of weathering. If would help the reader to list those (rather than to invite them to look up the paper themselves) and to indicate to what extent their review helps addressing these hypotheses. For instance hypothesis 8 (elevated $CO_2$ will enhance weathering) seems to assume that weathering fluxes and its ultimate consequences of drawdown of $CO_2$ occur at very different time scales, which could put constraints on feedback

mechanisms postulated in the ms. With respect to hypothesis 2, the importance of stoichiometry, I think that the studies done of mycorrhizal weathering provide much of the needed data. In none of their hypotheses they draw attention to different kinds of mycorrhizal symbiosis (but this could be a refinement of hypothesis 1), although it may not be coincidental that their figure 1 depicts an ectomycorrhizal conifer...

*A full discussion of all 12 hypotheses discussed by Brantley et al is not possible within this article is not possible for reasons of space but some of the ideas cited in that article are now discussed in more detail in as much as they relate to biological weathering.*

Some of these comments will make the manuscript longer, so I think it may help if I indicate cases were shortening of the ms is possible. I noted several digressions (also in the Abstract) that result in a less focused paper. Examples are: reference to acquisition of N and P by mechanisms other than weathering (p. 7, l. 18-34), hydraulic lift / redistribution (p. 9, l. 6-8), oxygenic and anoxygenic photosynthesis (p. 11, 14- 24), autotrophic and heterotrophic respiration in forests (p. 13, l. 30 – p. 14, l. 6; unless the authors think that Sverdrup is, essentially, right...), differential carbon storage in ectomycorrhizal and arbuscular mycorrhizal forests (p. 14, l. 4-28; note that of the two biomes where both guilds occur larger C storage per unit N was shown for the temperate biome, not for the (sub-)tropical biome – so we should not take Averill's claim too seriously), nitrogen in the rhizosphere (p. 24, l. 1-2).

*We will make some of the suggested cuts to reduce the overall length of the article.*

**Page-by-page comments**

**p. 3, l. 9**
Is the Finlay & Clemmensen paper on biogenic weathering? The title of the paper would suggest otherwise
*The paper is on carbon flow in relation to both decomposition and weathering*.

**p. 5, l. 14** Please provide a reference for the suggestion about the importance of horizontal gene transfer in such microbial consortia in EPS.
*Reference is now provided*.

**p. 7, l. 1**
Here I disagree. In such habitats, in case of a low pH, plants with cluster roots (or proteoid roots; I think they are the same) or dauciform roots produce carboxylates that *desorb* phosphorus from mineral surfaces. But desorption is not weathering, dissolution of minerals. Weathering would happen in the case of high-pH with calcium phosphates; in low-pH soils P is far too scarce to form substantial amounts of Fe- and Al-phosphates that are weathered.
*We agree with the reviewer and have re-written the text more carefully*

**p. 9, l. 14**
In the light of current criticisms of humic and fulvic acids as large molecules (Lehmann & Kleber claim these to be aggregates of essentially small molecules) this statement may need reconsideration in terms of underlying mechanisms.
*Agreed – we have altered the text to reflect this*

**p. 15, l. 21**

When introducing the Blum et al. hypothesis, they should also refer to contradictory data by Dijkstra and Smits (now only referred to on p. 17, l. 18-22; however I interpret that paper as showing that Blum et al.'s conclusion is grossly overstated – but I would love to see the opinion of the authors of this ms).

*Agreed – we have altered the text to reflect this. We originally discussed the Blum paper in more detail but removed the text because of this overstatement.*

**p. 17, l. 19**
Please provide a reference to that further study.

*The reference is now added – the comparable forest referred to is actually mentioned in the same study by Dijkstra & Smits.*

**p. 19, l. 17**
Note that exudation of carboxylates / organic anions can also have a major function in the desorption of iron-oxide bound soil organic matter and the subsequent acquisition of carbon (Keiluweit et al., Nature Clim. Change 5: 588-595. 2015) and nitrogen (Jilling et al., Biogeochemistry 139: 103-122. 2018) (and possibly phosphorus, as both inorganic and organic P are sorbed on such surfaces).

*Agreed – we have altered the text to reflect this*

Thomas W. Kuyper

---

## Author Comment (AC2) · 20 May 2019

**General comments**:

Finley and coauthors provide here an interesting and timely review on biological weathering across scales. It is well written and meets current questions and gaps of knowledge in this field. The general organization of the manuscript might on the other hand be significantly improved. I do not doubt however that some restructuring will enable this discussion paper to reach a wide audience and the large impact it deserves.

*We are grateful for these positive comments and agree that some re-structuring will improve the paper*

I would first like to acknowledge the fact that covering such a wide topic is chellenging, and I would like to congratulate the authors for their effort to try to bring together various aspects of the study of biological weathering in one single review paper. In that respect, I found the general organization according to spatial scales very attractive in the first place. The resulting sections, however, lack of focus, while the last sections do not seem to follow this original plan (e.g. section 6 on insights from stable isotope methods). As a result, the reader might get easily lost or distracted by some of the digressions.

*From the introduction (**section 1**) we review processes and experimental analyses at successively larger spatial and temporal scales (**sections 2,3,4 & 5**). Thereafter we discuss new possible methodological approaches using stable isotopes (**section 6**) and modelling (**section 7**) before presenting some concluding remarks (**section 8**) and finally (**section 9**) outlining some key questions, knowledge gaps and suggested future approaches. We think this structure is logical and will now underline*explain it at the beginning of the revised manuscript*so there is less likelihood of readers getting lost. **Section 9** will also be expanded.*

I think that the richness of ideas and concepts gathered here is a real originality of this review, but the author may want to be careful that the reader keeps track of the point that they are trying to make in a given paragraph. Section 5, which gathers a main section introducing concepts as diverse as "mineral evolution", the geological carbon cycle or plants as holobionts and another subsection on carbon allocation and sequestration including carbon cycle and geoengineering concepts is for instance a little hard to digest.

*We agree that that the review includes ideas from diverse disciplines (and also that this contributes to the originality of the review) but we also accept that these ideas can be introduced in a way that makes them more "digestible" and we have tried to do that in the revised manuscript.*

To improve this point, I could first suggest gathering the different processes and links existing between them in a dedicated introductory section to make sure all readers are on the same page before tackling more detailed aspects of each scale. For instance, the relevance of allusions to long-term sequestration of carbon (e.g. lines 7-9 p. 8 and lines 24-26 p. 6) for the general topic of the paper might be unclear to some readers until they reach section 5.1. Another example is the

geological cycle of carbon, the presentation of which is scattered across section 5 and somewhat redundant (e.g. p. 11 and 14). An introductory section could also enable to present the order of magnitude of the different processes and elemental fluxes to be considered here (e.g. typical elemental flux derived from primary mineral weathering vs. typical plant uptake and po- tential export related to forestry practices vs. typical atmospheric input for a given type of system) which is something missing here. Second, I would recommend organizing sections into subsections to keep the reader oriented. I would also avoid sections including a sort of single small subsection, e.g. 5->5.1->6 or 7->7.1->8.

*We agree with these helpful suggestions and will re-write the introduction to describe the structure of the article and introduce the keep concepts to be discussed. We will change the section divisions to make them more consistent between chapters and try avoid single small subsections.*

Another general point is that I find that the manuscript is lacking a few but quite important references. I try to provide a couple of them in the specific comments section below, which I hope the authors will find helpful. Aside from those points, I am enthusiastic about this interesting manuscript and I would recommend its publication provided that a couple of modifications and restructuring are done.

*We are pleased this reviewer recommends publication, welcome the suggestions concerning re-structuring and will try to follow the helpful advice to make the article clearer.*

**Specific comments:**

**-Section 2**: Alt and Mata (2000), Benzerara et al. (2007), Furnes et al. (2001) and Torsvik et al. (1998) are additional references on the biotic origin of tubular structures that the authors might find useful to include. l.12 p.4: the effect of turgor pressure on biomineral weathering is also discussed by Li et al. (2016)

*Thanks for these helpful suggestions*

**-Section 3**: Maybe the first paragraph might be strengthened by adding a couple of references when presenting common biofilm features to guide the reader, especially if some studies are relating these biofilm properties (e.g. retention of water) to mineral weathering (e.g. fluid-mineral contact time). In the second paragraph, Barker et al. (1998) is probably another classical reference on biofilms and microenvironments that might be added. In the last paragraph dealing with the interplay between bacteria and mineral weathering should be strengthened in my opinion. Some recent references including Mitchell et al. (2013), Montross et al. (2013), Wild et al. (2018) and Wild et al. (2019) are missing here and should be included at this point I think. l.19 p.6: "Burial" is referred to as "incubation" in Uroz et al. (2012). I would recommend sticking to this latter term. l.23 p.6: I am not completely sure of the relevance of the position of the last sentence (l. 23-26). I would move it upward or delete it.

*Thanks for these helpful suggestions*

**-Section 4**: l.29 p.6: the statements of the production of acidifying substances (H+, organic acids) and ligands that complex with metals in the minerals may need to be supported by quotations. l.30 p.6:: "that retard weathering rates" reduce or decrease weathering rates would be more accurate l.7 p.7: "uptake of positively charged nutrients such as NH4+ and K+, result in exudation of protons" may benefit from the support of a quotation. *OK*

**-Section 5**: This section is a little bit dense, I would suggest dividing it into subsections. C3

*Agreed – we will use sub-section titles to improve readability.*

**-Section 6**: This section is thematic, not intrinsically associated to a given scale. Also, I am questioning the scientific relevance of specifically distinguishing studies from the QWARTS project from other studies.

*Hopefully the re-structuring of the introduction will improve the readability and make it easier to understand our approach. Theoretically stable isotope measurements could be discussed within each scale section but we thought it was easier to group these studies together – especially as we introduced new results from a hitherto unpublished study. The fact that these results are so far un-published is one reason for distinguishing them and the idea behind the special issue was to highlight the recent research done within this interdisciplinary project but we can remove the reference to the project if the Editor thinks this is more appropriate.*

-**Section 7**: Direct in situ measurements using gravimetric approaches by Augusto et al. (2000) or Turpault et al. (2009) or interferometry methods by Wild et al. (2019) are not reported by Akselsson et al. (2019) but might be worth mentioning since they directly meet some of the challenges implicitly pointed out in this manuscript regarding the validation of weathering models and the transposition/upscaling of laboratory mesocosms to field systems. In the second paragraph, I find the description of the influence of the different processes on the dissolution rate a little bit unclear, and I feel that the clarity of this section might be improved. Otherwise, readers who are not familiar with that type of models will be easily lost. I would suggest reorganizing this section and starting by presenting the different parameters controlling the dissolution rate (temperature, pH, chemical affinity, . . .) and then, in a second step, describing the influence of plant metabolism on these factors and thus on the dissolution rate. I would also strongly recommend using an equation (e.g. developed from equation 3 in Erlandsson et al. (2016), equation 3 in Godderis et al. (2006) or equation 1 in Palandri and Kharaka (2004)) to visually support this discussion. I would also avoid mentioning the concepts of "weathering brakes" or "transition state theory" if they are not explained. This might be more confusing than useful for readers, depending on their background.

*We thank the reviewer for these detailed, helpful suggestions and will do our best to incorporate them in our revised manuscript. We aim to shorten section 7 substantially and to re-structure and simplify it so that it is more directly relevant to the weathering processes described in the rest of the manuscript.*

*Thank you for these additional references*

**References:**

Alt, J.C., Mata, P., 2000. On the role of microbes in the alteration of submarine basaltic glass: a TEM study. Earth and Planetary Science Letters 181, 301-313.

Augusto, L., Turpault, M.P., Ranger, J., 2000. Impact of forest tree species on feldspar weathering rates. Geoderma 96, 215-237.

Barker, W.W., Welch, S.A., Chu, S., Banfield, J.F., 1998. Experimental observations of the effects of bacteria on aluminosilicate weathering. Am. Miner. 83, 1551-1563.

Benzerara, K., Menguy, N., Banerjee, N.R., Tyliszczak, T., Brown, G.E., Guyot, F., 2007. Alteration of submarine basaltic glass from the Ontong Java Plateau: A STXM and TEM study. Earth and Planetary Science Letters 260, 187-200.

Erlandsson, M., Oelkers, E.H., Bishop, K., Sverdrup, H., Belyazid, S., Ledesma, J.L.J., Köhler, S.J., 2016. Spatial and temporal variations of base cation release from chemi- cal weathering on a hillslope scale. Chem. Geol. 441, 1-13.

Furnes, H., Muehlenbachs, K., Torsvik, T., Thorseth, I.H., Tumyr, O., 2001. Microbial fractionation of carbon isotopes in altered basaltic glass from the Atlantic Ocean, Lau Basin and Costa Rica Rift. Chem. Geol. 173, 313-330.

Godderis, Y., Francois, L.M., Probst, A., Schott, J., Moncoulon, D., Labat, D., Viville, D., 2006. Modelling weathering processes at the catchment scale: The WITCH numerical model. Geochim. Cosmochim. Acta 70, 1128-1147.

Li, Z.B., Liu, L.W., Chen, J., Teng, H.H., 2016. Cellular dissolution at hypha- and spore- mineral interfaces revealing unrecognized mechanisms and scales of fungal weather- ing. Geology 44, 319-322.

Mitchell, A.C., Lafreniere, M.J., Skidmore, M.L., Boyd, E.S., 2013. Influence of bedrock mineral composition on microbial diversity in a subglacial environment. Geology 41, 855-858.

Montross, S.N., Skidmore, M., Tranter, M., Kivimaki, A.L., Parkes, R.J., 2013. A micro- bial driver of chemical weathering in glaciated systems. Geology 41, 215-218.

Palandri, J.L., Kharaka, Y.K., 2004. A compilation of rate parameters of water-mineral interaction kinetics for application to geochemical modeling, in: Survey, U.S.G. (Ed.), U.S. Geological Survey, Open File Report. U.S. Geological Survey, Open File Report, p. 70.

Torsvik, T., Furnes, H., Muehlenbachs, K., Thorseth, I.H., Tumyr, O., 1998. Evidence for microbial activity at the glass-alteration interface in oceanic basalts. Earth and Planetary Science Letters 162, 165-176.

Turpault, M.-P., Nys, C., Calvaruso, C., 2009. Rhizosphere impact on the dissolution of test minerals in a forest ecosystem. Geoderma 153, 147-154.

Uroz, S., Turpault, M.P., Delaruelle, C., Mareschal, L., Pierrat, J.C., Frey-Klett, P., 2012. Minerals affect the specific diversity of forest soil bacterial communities. Geomicrobiol- ogy Journal 29, 88-98.

Wild, B., Daval, D., Beaulieu, E., Pierret, M.-C., Viville, D., Imfeld, G., 2019. In-situ dissolution rates of silicate minerals and associated bacterial communities in the critical zone (Strengbach catchment, France). Geochim. Cosmochim. Acta 249, 95-120.

Wild, B., Imfeld, G., Guyot, F., Daval, D., 2018. Early stages of bacterial community adaptation to silicate aging. Geology 46, 555-558.

---

## Author Comment (AC3) · 20 May 2019

**In General:**

The authors of the paper propose a review of current (last 10 years) of advancement made in the understanding of biological weathering, specifically focusing on the boreal forest, in response to an interdisciplinary project called "Quantifying weathering rates for sustainable forestry." This topic fits well the scope of BG and it is needed to help us move forward in this area of research.

*Good that the topic fits and that an article like this is needed.*

The manuscript cites more than 200 references that span a wide range of topics from physical, chemical and biological approaches, and scales from nano-scale empirical studies to global scale modeling, and emphasizing an evolutionary viewpoint on biological weathering. However, I was disappointed seeing that about 1/3rd of the references are prior to 2009 and have been widely cited and reviewed in the past, thus these do not give "anything new" especially in some sections of the manuscript (see details below) and it also contradicts with the authors aim of summarizing the last 10 years of advancement.

*We understand this reasoning but inclusion of some older references is necessary to provide perspective and to explain the development of different types of experiments. We will try to reduce the proportion of older references by cutting some of the older ones.*

The manuscript is a well-written review/summary of more than the last 10 years of biological weathering research with a heavy emphasis on mycorrhizae mediated weathering (which is the ecosystem in the boreal forest). However, there is no synthesis of the reviewed literature, there is no agreement stated on what is the current understanding, or state of this biological weathering in the boreal forest, and how it applies to sustain- able forestry or simply how to move forward. The manuscript is a review, but it lacks a synthesis.

*We agree that the aims of the article are not clearly stated and that the final take home message is not made clearly enough. We will try to provide the desired synthesis more clearly by re-structuring of the manuscript.*

Regardless of great writing, it was not an easy read, because I could not find/follow the purpose of this manuscript, it presents a lot of data on both side of the arguments that contradict each other, which is fine, however, there are no directions, there is a lot of rambling on without focus – what is the underlying message? What do the authors want to achieve with this review? Key questions and knowledge gaps section is underdeveloped and it seems like it was an afterthought and stuck to the end.

*We are pleased that the reviewer considers the writing was good but agree the take-home message was not stated clearly enough. In the revision we will address this problem and try to provide more focus. We agree that the last section is too superficial and under-developed. It was included too close to the initial deadline and will be expanded to provide clearer guidance about key knowledge gaps and necessary approaches to resolving conflicting opinions in future experiments.*

In addition, I think that the title is misleading, as the review is really about weathering in the boreal forest. Most cited work was done by researchers related to the boreal or other forests (field), in the laboratory using mostly conifers and mycorrhizal fungi, and there are couple of "side topics" that seems to be out of place in this bigger scheme (for example, the hydraulic lift study for drought-prone ecosystems).

*We agree that the main emphasis is on boreal forests, although the section on evolutionary aspects includes a discussion of processes that took place before the evolution of terrestrial plants. Other components of different ecosystems are also mentioned in this section including 1. proteoid roots of in highly weathered soils, 2. calcicole plants in calcareous soils, 3. non-mycorrhizal fungi such as different Aspergillus species and 4. different bacterial species. If the handling editor considers it appropriate we can add a secondary part to the title such as "- with particular emphasis on boreal forests."*

**Some specifics:**
**Abstract and 1. Introduction** – no specific comments.
**2. Microscale/nanoscale observations of physical alteration of minerals:** This section is heavily based on older findings and mention some new studies, but it is unclear what advances were made in the last 10 years – new techniques? New understanding of processes? Or just supporting previous findings? Or all above? It needs a refocus, and it can be shortened by about half and still convey the same message.

*We agree, in part, with this assessment and will try to emphasize more recent studies involving the application of new techniques, and to explain more clearly what advances have been made – where appropriate. This section is only about 880 words (6.8 % of the article) and not excessively long but we will attempt to reduce the length.*

**3. Biofilms and small-scale microbial interactions** with consequences at higher spatial scale: how are these differently categorized than the next section, which is about microbial and plant secretions? EPS, biofilm, oxalic acids etc. are secretions, are not? What are the consequences at higher spatial scale? Do we know? Or is it a challenge to scale things up? Again, what is the new advancement in the last 10 years? The section needs some clarifications and/or refocus.

*The interactions in this section take place at a smaller spatial scale than those discussed in the subsequent section where plants or microorganisms are cultured in micro- or mesocosms. Admittedly there is some overlap between these sections since these small-scale processes also take place in single plant-scale interactions studied in microcosms, but we will re-write to improve clarity and focus.*

**4. Microbial and plant secretions** – evidence from microcosms and mesocosms: long section – rambling on without focus, lots of info and data about various roles, functions, and processes of mycorrhizal fungi, but no other components of the ecosystem, and the hydraulic lift section seems irrelevant in the boreal forest. Bringing in drought may be something we want to think about as climate shifts, but it most likely causing larger problems in drought-prone parts of the world.

*We disagree about other components not being mentioned in this section. Other components of (different) ecosystems **ARE** mentioned in this section including **1.** proteoid roots of plants growing in highly weathered soils, **2.** calcicole plants growing in calcareous soils, **3.** non-mycorrhizal fungi such as different Aspergillus species and **4.** different bacterial species. We will try to add some more comments about other non-mycorrhizal fungi and lichens. We will*

*also try to reduce the rambling and improve focus. We can remove the reference to hydraulic lift if necessary.*

**5. Systemic consequences of microorganism-mineral interactions in an ecological and evolutionary context:** this is really important and interesting, however, it is too long, have some repetition – I am not sure why the 5.1. section is separated (elevated) from the rest of 5. – Weathering, nutrient acquisition, carbon allocation, and sequestration are the key elements of the evolutionary viewpoint – perhaps, this section could be rearranged and shortened to synthesize our current understanding of the evolution of plants and associated fungi in the context of carbon and nutrient cycling. Bob Berner did the pioneering work in this field with his carbon models, but it got a lot of attention in the last 10 years, so a focused synthesis would help us to identify future directions.

*We agree that this section can be made more concise and can be re-arranged. We want to retain the evolutionary focus to underline the fundamental nature of the interactions between microorganisms and minerals but we can provide more of a focused synthesis about future directions of research and outstanding questions.*

**6. Methods using stable isotopes:** The section is interesting, provide laboratory evidence of the usefulness of these techniques in addition to field studies, however, the last paragraph states that the "there is no clear evidence that processes observed at the laboratory-scale play a significant role in "soil-scale" mineral dissolution rates." This indicates that laboratory studies are useless, why do we bother then? Is there anything we learned from the laboratory studies? Also, the last paragraph is a repetition of statements on page 12 lines 13-15.

*We have agreed to remove this statement from the last paragraph. We will also re-write this section to avoid any repetition.*

**7. Modelling of weathering in forest soils:** this whole section is unfocused. It starts with the PROFILE and ForSAFE models, then it talks about information needs and possible improvements (in 7.1.) and then it returns to talk about a bunch of other models in too much detail without getting to a point. This section should synthesize what are the main outcomes of the different modeling approaches (probably in half of the length), and identify what is missing (information) and how to tackle the shortcomings.

*We agree with these comments and will re-write this section according to the recommendations. It will be re-organized and shorted substantially.*

**8. Conclusions:** I was expecting to find the key questions, knowledge gaps and future directions (or call for specific areas of research) in this section.

*We agree that this section could be usefully combined with Section 9 (which is much too short). We will expand the key conclusions and have a clear presentation of the major knowledge gaps as well as clear recommendations about how these can be solved. – To identify the key questions and what future approaches/measurements are needed to answer them.*

**Figures:** Not all necessary – Figure 1, 2, 4, 6, 7 do not add new information to the summary (synthesis) or not necessary to understand the text. Figure 9 and 10 are a good representation of specific examples for laboratory approaches. Figure 3, 5, 8, and 11 are great illustrations of processes and their interactions from small to large scales.

*We agree with this assessment of the necessity (or not) of the figures. The non-essential ones were provided to make the article more self-contained but can be omitted to make the article shorter. Figures 4 and 6 at least show what mycorrhizas look like and show a pictorial representation of what they do (in terms of C allocation). Potentially the article may be read by many people who have limited knowledge of mycorrhizal structures and these would improve understanding.*

---

## Author Comment (AC4) · 20 May 2019

We would like to thank all three reviewers for providing helpful, constructive comments that will help us to provide a more concise, clearer, revised manuscript. All three reviewers seem to feel that the manuscript is well written but it is clear that parts of it are too long and that, although it is a good "review" we need to try harder to synthesize the information to provide a clearer take home message. In particular we need to develop the final sections to provide better guidance about future research approaches and questions. We hope to be able to achieve all these goals if we are allowed to submit a revised manuscript.

---

## Author Response (AR1)

**Biological weathering and its consequences at different spatial levels – from nanoscale to global scale**

Roger D. Finlay[1], Shahid Mahmood[1], Nicholas Rosenstock[2], Emile B. Bolou-Bi[3], Stephan J. Köhler[4], Zaenab Fahad[1,5], Anna Rosling[5], Håkan Wallander[2], Salim Belyazid[6], Kevin Bishop[4] and Bin Lian[7]

[1]Uppsala BioCenter, Department of Forest Mycology and Plant Pathology, Swedish University of Agricultural Sciences, SE-750 07 Uppsala, Sweden
[2]Department of Biology, Lund University, Box 117, SE-221 00 Lund, Sweden
[3]Université Felix Houphouët-Boigny, UFR des Sciences de la Terre et des Ressources Minières, Departement des Sciences du sol, BP 582 Abidjan 22, Côte D'Ivoire
[4]Soil-Water-Environment Center, Department of Aquatic Sciences and Assessment, Swedish University of Agricultural Sciences, SE-750 07 Uppsala, Sweden
[5]Department of Ecology and Genetics, EBC, Uppsala University SE-752 36 Uppsala, Sweden
[6]Department of Physical Geography, Stockholm University, SE-106 91 Stockholm, Sweden
[7]College of Life Sciences, Nanjing Normal University, Nanjing 210023, China

*Correspondence to*: Roger D. Finlay (Roger.Finlay@slu.se)

**Authors' responses**

Detailed replies to the reviewers' comments have already been posted during the interactive discussion. Below we add additional comments.

The major criticism of the manuscript was that, while it was a good review, it did not function as a *synthesis*. We have attempted to rectify this by being clearer about our own conclusions and by providing a clearer distinction between **1**) work that was performed before 2009, **2**) work that has been carried out in the following 10 years and **3**) experiments that still need to be performed (where there are still knowledge gaps).

We have added a summary section containing this information at the end of each of the first six sections. We hope that the overall aim of the review is now clearer. Almost all of the recommendations we received from the referees have been acted upon. A number of addition references suggested by reviewer 1 have been added. There is some additional text on lichens and saprotrophic fungi.

Reviewer 2 considered the proportion of older references was too high so we have removed about 25 of the older references and replaced them with more modern work. Several of the submitted manuscripts we referred to are now published.

We have followed the recommendation of Reviewer 2 and removed five of the original 11 figures and there are now six. Some of the missing references suggested by Reviewer 3 have been added but most of the older ones were not – bearing in mind the criticism of Reviewer 2. The manuscript is still quite long, but this is necessary since it covers a broad range of fields. We hope that the re-ordering of the figures will make the manuscript easier to follow. Five sections deal with different spatial scales. The sixth and seventh sections deal with stable isotope experiments and modelling respectively – two areas where further advances can be made in understanding.

In the conclusion we mention the important conclusions of Severdrup et al (2009) referred to by reviewer 1 under his point 5.

We hope the manuscript is significantly improved now and that it will soon be acceptable for publication in Biogeosciences.

Roger Finlay.

[revised manuscript text omitted]

---

## Author Response (AR2)

Natascha Töpfer
Copernicus Publications
Editorial Support
editorial@copernicus.org

Dear Editor

Thank you for the comments of the associate editor on our manuscript **bg-2019-41.**

We agree that section 5 had become a little long, following incorporation of the material that the referees considered we should mention. We have followed the recommendation of the associate editor and reduced the length of this section by
10%. We have also made a number of other small changes and deletions in other sections of the manuscript and tried to improve the clarity of the text in several places.

The changes are listed below in this document and we have uploaded a clean version of the revised manuscript.

We hope that the revised version is more suitable for publication.

With best regards,

Roger Finlay.

[revised manuscript text omitted]

---

## Author Response (AR5)

**Response to reviewer's comments on bg-2019-41**

We were pleased that the referee appears to have accepted our response to his earlier comments and that he has only suggested a number of minor revisions – principally possible ways to shorten the manuscript.

We have adopted many, but not all, of the reviewer's suggestions and the revised manuscript has been shortened by a further 7%. Comments on the changes made are shown below and in the revised text in this document. Please note that the page numbers referred to by the reviewer are those in our original response document and are not the same as the page numbers in this document.

No changes are required to the figures that were previously submitted to the production department of the journal.

**Suggestions for revision or reasons for rejection (will be published if the paper is accepted for final publication)**

In this revision that authors have attempted to shorten the ms, which admittedly had become too lengthy because of incorporation of suggestions by reviewers. I would agree that all deletions performed did not reduce the information contained in the ms.
In fact, when reading this version I thought that there were quite a few further options for reductions. Specifically this pertains to the various paragraphs that deal with acquisition of N – for which the link to weathering is remote at best. Specifically this easy shortening refers to p. 9, l. 18-23; p 11, l. 13-17, and 22-28; p. 17, l. 17-28; p. 20, l. 23-28 [not about isotopes, the subject of that section, and largely repetitive]; p. 22, l. 31-32; p. 27, l. 11-20. The section of past mycorrhizal symbioses in the framework of the evolution of the oxygen and carbon dioxide cycles (p. 13-14) also seems somewhat out of place in this paper on weathering. Finally I think a substantial reduction can be achieved by rearranging section 7. If the authors starts with section 7.2 (the models), they can then combine section 7.1 and 7.3 (which parameters should be included in the model and for which parameters do we need better estimates?) and discover that after the current section 7.1, section 7.3 adds little novelty but only makes general statements.

I would urge the authors to be consistent in terminology, which is now somewhat inconsistent possibly due to the fact that the lead author did not impose a uniform terminology. What are deprotonated organic acids on p. 16 seem to be the same as organic ligands in section 7. Considering the debates surrounding weathering mechanisms (acidification on the one hand and / or ligand formation by organic anions /carboxylates on the other), I think it beneficial that, if possible, both classes of mechanisms are kept separate; and there a term like deprotonated acid is a poor expression.
If apatite needs a further explanation (but considering that the other minerals are not explained, one wonders whether this is needed for apatite), it is best to do so at first mention. After te word has been used 14 times, readers would likely not be in need any more to learn that apatite is a calcium phosphate mineral).

**Commented [RF1]:** POINT 1. Blue text about relevance of N acquisition. This link is not "remote". If plants cannot acquire enough N to grow and allocate C to mineral substrates they cannot weather minerals. This is illustrated dramatically in **Figure 6** and is a major point of relevance to sustainable forestry.

**Commented [RF2]:** POINT 2. **P9, l. 18-23.** This text is not about N – it is about acquisition of P in relation to changes in atmospheric $CO_2$.

**Commented [RF3]:** POINT 3. **P11, l. 13-17 & 22-28.** The (partial) de-coupling of microbial activity from changes in soil moisture, and the potential role of sugars in polyols in generating turgor pressure in fungal hyphae are both factors of direct importance to weathering by fungi. **However, the second section has been deleted according to the reviewer's suggestion.**

**Commented [RF4]:** POINT 4. **P17, l. 17-28.** This section has been re-written to make it shorter. The references to global drawdown of $CO_2$ are important and central to this article but **we have removed the second half of this section (Lines 24-28) to make the text shorter and reduce overlap with section 7.**

**Commented [RF5]:** POINT 5. **P20, l. 23-28** This section **IS** about isotopes and discusses results presented in **Fig 5** and discussed in the following sentence. The first sentence presents relevant (contrasting) results obtained using X-ray diffraction and we wish to retain this text.

**Commented [RF6]:** POINT 6. **P22, l. 31-32.** This is a comment about how weathering and N acquisition **ARE** connected. It is well accepted that plant growth affects weathering and that N acquisition affects the growth of trees. This indirect connection between weathering and N acquisition (from organic matter) is demonstrated in **Figure 6**.

**Commented [RF7]:** POINT 7. "past mycorrhizal symbioses" are *not mentioned at all* in relation to the evolution of oxygen. The "Great Oxidation Event" took place over **two billion years before ectomycorrhizas evolved** and involved other microorganisms. The effects of weathering on possible $CO_2$ drawdown discussed later on page 14 in relation to the rise of land plants and development of forests are highly relevant to the effects of weathering. The purpose of this section is to discuss the importance of microbe-mineral interactions (biological weathering) and the systemic consequences of this on an evolutionary scale – before and during the evolution of plants.

**Commented [RF8]:** POINT 8. **This section on modelling has been revised in accordance with the reviewer's suggestions and is now about 25% shorter**

**Commented [RF9]:** POINT 9. **We have revised the terminology according to the reviewer's suggestions – to make it more consistent.**

**Commented [RF10]:** POINT 10. The reason calcium phosphate was mentioned specifically in this section is that we were discussing possible isotopic tracers. However **we have now re-written this section to remove the term.**

[revised manuscript text omitted]